EMBO
Molecular Medicine

# RGS9-2 rescues dopamine D2 receptor levels and signaling in *DYT1* dystonia mouse models

Paola Bonsi[1,*] , Giulia Ponterio[1,2], Valentina Vanni[1,2], Annalisa Tassone[1,2], Giuseppe Sciamanna[1,2], Sara Migliarini[3], Giuseppina Martella[1,2], Maria Meringolo[1,2], Benjamin Dehay[4,5] , Evelyne Doudnikoff[4,5], Venetia Zachariou[6], Rose E Goodchild[7] , Nicola B Mercuri[1,2], Marcello D'Amelio[8,9], Massimo Pasqualetti[3,10], Erwan Bezard[4,5] & Antonio Pisani[1,2,**]

## Abstract

Dopamine D2 receptor signaling is central for striatal function and movement, while abnormal activity is associated with neurological disorders including the severe early-onset *DYT1* dystonia. Nevertheless, the mechanisms that regulate D2 receptor signaling in health and disease remain poorly understood. Here, we identify a reduced D2 receptor binding, paralleled by an abrupt reduction in receptor protein level, in the striatum of juvenile *Dyt1* mice. This occurs through increased lysosomal degradation, controlled by competition between β-arrestin 2 and D2 receptor binding proteins. Accordingly, we found lower levels of striatal RGS9-2 and spinophilin. Further, we show that genetic depletion of RGS9-2 mimics the D2 receptor loss of *DYT1* dystonia striatum, whereas RGS9-2 overexpression rescues both receptor levels and electrophysiological responses in *Dyt1* striatal neurons. This work uncovers the molecular mechanism underlying D2 receptor downregulation in *Dyt1* mice and in turn explains why dopaminergic drugs lack efficacy in *DYT1* patients despite significant evidence for striatal D2 receptor dysfunction. Our data also open up novel avenues for disease-modifying therapeutics to this incurable neurological disorder.

**Keywords** beta-arrestin; lysosomal degradation; striatum
**Subject Categories** Neuroscience; Pharmacology & Drug Discovery

## Introduction

Striatal dopaminergic transmission is central to movement control and several disease conditions (Redgrave *et al*, 2010; Gittis & Kreitzer, 2012). Dopaminergic dysfunction has been implicated in early-onset generalized *DYT1-TOR1A* dystonia, a highly disabling and incurable neurological disease typically manifesting in childhood, which generalizes within a few years causing involuntary movements and abnormal postures (Balint *et al*, 2018). This disorder is most frequently caused by an autosomal dominant Δgag mutation in the *TOR1A* gene, causing loss of function of the gene product torsinA, a member of the AAA+ (ATPases associated with cellular activities) family of proteins (Ozelius *et al*, 1997). Mutant ΔE-torsinA is mislocalized from the endoplasmic reticulum to the nuclear envelope, causing abnormalities in folding, assembly, and trafficking of proteins targeted for secretion or to membranes (Torres *et al*, 2004; Burdette *et al*, 2010; Granata *et al*, 2011).

Clinical neuroimaging studies have revealed decreased caudate–putamen dopamine D2 receptor (DRD2) availability in *DYT1* patients compared to controls (Asanuma *et al*, 2005; Carbon *et al*, 2009). Reduced striatal DRD2 binding and protein level have also been reported in several different *DYT1* experimental models (Napolitano *et al*, 2010; Yokoi *et al*, 2011; Dang *et al*, 2012). Notably, multiple lines of evidence demonstrated reduced coupling between the DRD2 and its cognate G proteins and severely altered receptor function (Pisani *et al*, 2006; Napolitano *et al*, 2010; Sciamanna *et al*, 2011, 2012a; Martella *et al*, 2014; Scarduzio *et al*, 2017).

DRD2 signaling *via* G proteins inhibits cAMP production and in turn PKA activity. However, there is also accumulating evidence for G protein-independent DRD2 signaling functions (Beaulieu &

1   Laboratory of Neurophysiology and Plasticity, IRCCS Fondazione Santa Lucia, Rome, Italy
2   Department of Systems Medicine, University Tor Vergata, Rome, Italy
3   Unit of Cell and Developmental Biology, Department of Biology, University of Pisa, Pisa, Italy
4   Université de Bordeaux, Institut des Maladies Neurodégénératives, UMR 5293, Bordeaux, France
5   CNRS, Institut des Maladies Neurodégénératives, UMR 5293, Bordeaux, France
6   Department of Neuroscience, Friedman Brain Institute, Icahn School of Medicine at Mount Sinai, New York, NY, USA
7   Department of Neurosciences, VIB-KU Leuven Center for Brain and Disease Research, KU Leuven, Leuven, Belgium
8   Laboratory Molecular Neurosciences, IRCCS Fondazione Santa Lucia, Rome, Italy
9   Unit of Molecular Neurosciences, Department of Medicine, University Campus-Biomedico, Rome, Italy
10  Center for Neuroscience and Cognitive Systems @UniTn, Istituto Italiano di Tecnologia, Rovereto, Italy
    *Corresponding author. Tel: +39-06501703211; E-mail: p.bonsi@hsantalucia.it
    **Corresponding author. Tel: +39-06501703153; E-mail: pisani@uniroma2.it

Gainetdinov, 2011), as well as G protein-independent regulation of the GPCR activity of DRD2. The reciprocal interactions of DRD2 with spinophilin or arrestin represent a regulatory mechanism for fine-tuning receptor-mediated signaling. Indeed, β-arrestin 2 (β-Arr2) is involved in internalization and G protein-independent signaling of DRD2 (Beaulieu *et al*, 2011; Del'guidice *et al*, 2011), while spinophilin antagonizes arrestin actions (Wang *et al*, 2004). In addition, the striatal-enriched regulator of G protein signaling 9-2 (RGS9-2) regulates the amplitude of the behavioral responses to DRD2 activation (Rahman *et al*, 2003; Gold *et al*, 2007; Traynor *et al*, 2009), inhibits DRD2 internalization (Celver *et al*, 2010), and specifically modulates DRD2 signaling in striatal neuronal subtypes (Cabrera-Vera *et al*, 2004). On the other hand, the receptor can target the RGS protein to the plasma membrane (Kovoor *et al*, 2005; Celver *et al*, 2012), and exposure to DRD2 ligands can alter RGS9-2 level in wild-type animals (Seeman *et al*, 2007) indicating a reciprocal modulation.

In the present work, we investigated the molecular mechanisms underlying DRD2 reduced levels and altered signaling in the striatum of *DYT1* dystonia models, $Tor1a^{+/-}$-knock-out and $Tor1a^{\Delta gag/+}$-knock-in mice (Goodchild *et al*, 2005). Our findings shed new light on DRD2 dysfunction in *DYT1* striatum and show that *in vivo* delivery of RGS9-2 is able to rescue DRD2 expression levels and to recover striatal D2DR signaling. These findings might explain the paradox of the lack of efficacy of dopaminergic drugs in *DYT1-TOR1A* dystonia patients, despite strong evidence that abnormal dopamine signaling is central to disease pathophysiology. Further, they also define a potential therapeutic target that restores dopaminergic responses.

## Results

### DRD2 and RGS9-2 protein levels are simultaneously downregulated in *DYT1* striatum

In order to analyze the molecular mechanisms of DRD2 dysfunction, we utilized the $Tor1a^{+/-}$ mouse model that mimics the loss of function effect of the *DYT1* dystonia *TOR1A* mutation.

First, we measured receptor expression levels in the striatum of adult (P60–P90) $Tor1a^{+/+}$ (Fig EV1) and $Tor1a^{+/-}$ male littermates. $Tor1a^{+/-}$ samples showed, as expected, lower amounts of torsinA protein (Fig 1A$_1$; $Tor1a^{+/+}$ 1.000 ± 0.053 N = 7, $Tor1a^{+/-}$ 0.602 ± 0.055 N = 6, *t*-test P = 0.0003). In lysates of $Tor1a^{+/-}$ striatum, we also observed significantly reduced levels of DRD2 protein compared to $Tor1a^{+/+}$ control samples (Fig 1A$_1$; $Tor1a^{+/+}$ 1.000 ± 0.027 N = 18, $Tor1a^{+/-}$ 0.839 ± 0.044 N = 16, *t*-test P = 0.0029), despite similar mRNA levels (Fig EV2; $2^{-dCt}$: $Tor1a^{+/+}$ 0.649 ± 0.081 N = 3, $Tor1a^{+/-}$ 0.587 ± 0.175 N = 3; Mann–Whitney test P = 1.0000). Based on the well-characterized reciprocal regulatory relationship between GPCRs and regulator of G protein signaling (RGS) proteins, we next examined whether RGS protein levels were affected and might give further insight on the nature of the impairment of D2R-mediated transmission in *DYT1* mutant mice. We focused this analysis on striatal levels of RGS9-2, an R7 RGS family member specifically regulating DRD2 function, and the closely related RGS7. Western blotting (WB) analysis revealed significantly reduced RGS9-2 levels in $Tor1a^{+/-}$ mice (Fig 1A$_1$; $Tor1a^{+/+}$ 1.000 ± 0.036 N = 32, $Tor1a^{+/-}$ 0.844 ± 0.049 N = 33, *t*-test P = 0.0115). Conversely, RGS7 levels were comparable between genotypes (Fig 1A$_1$; $Tor1a^{+/+}$ 1.000 ± 0.077 N = 7; $Tor1a^{+/-}$ 0.980 ± 0.097 N = 7, *t*-test P = 0.8721), ruling out a generalized protein downregulation and pointing to a specific deficit of DRD2 and its signaling pathway.

To assess the reciprocal relationship among these dysregulated proteins, we then analyzed the time-course of striatal expression levels of torsinA, DRD2, and RGS9-2 during postnatal development (P7, P14, P21, P60). TorsinA levels (Fig 1A$_2$) were significantly reduced in $Tor1a^{+/-}$ with respect to control mice at P7, P14 and P60 (one-way ANOVA P < 0.0001 and Bonferroni's multiple comparison test P < 0.01). In contrast, DRD2 levels increased similarly between P7 and P21 in both genotypes (Fig 1A$_3$; $Tor1a^{+/+}$: P7 0.404 ± 0.169, P21 0.991 ± 0.072; $Tor1a^{+/-}$: P7 0.595 ± 0.140, P21 0.980 ± 0.111; one-way ANOVA P = 0.0023 and Bonferroni's multiple comparison test: $Tor1a^{+/+}$ vs. $Tor1a^{+/-}$ P > 0.05), consistent with the previously reported developmental progression of mRNA levels in the murine striatum (Araki *et al*,

---

**Figure 1. Striatal levels of DRD2 and RGS9-2 are reduced in the $Tor1a^{+/-}$ and $Tor1a^{\Delta gag/+}$ DYT1 dystonia mouse models.**

A    (A1) Representative WBs showing DRD2 and RGS9-2 downregulation in adult (P60–P90) $Tor1a^{+/-}$ (+/−) striata, characterized by a reduced torsinA protein level with respect to $Tor1a^{+/+}$ (+/+) littermates. The striatal level of RGS7 is unchanged. The dot plot shows $Tor1a^{+/-}$ data, normalized to $Tor1a^{+/+}$ controls of the same experiment. TorsinA: $Tor1a^{+/+}$ N = 7, $Tor1a^{+/-}$ N = 6, *t*-test ***P = 0.0003; DRD2: $Tor1a^{+/+}$ N = 18, $Tor1a^{+/-}$ N = 16, *t*-test **P = 0.0029; RGS9-2: $Tor1a^{+/+}$ N = 32, $Tor1a^{+/-}$ N = 33, *t*-test *P = 0.0115; RGS7: $Tor1a^{+/+}$ N = 7, $Tor1a^{+/-}$ N = 7, *t*-test P = 0.8721. WB quantification data, expressed as the ratio of protein vs. loading control intensity level, are normalized to the wild-type samples of the same experiment. (A2–A4) Time-course of changes in torsinA, DRD2, and RGS9-2 striatal levels along postnatal development. Summary plot data are normalized to the $Tor1a^{+/+}$ P60 sample of each independent experiment. (A2) TorsinA level is significantly reduced in mutants throughout the developmental period considered (P7 N = 3; P14–P60 N = 4; one-way ANOVA P < 0.0001 and Bonferroni's multiple comparison test **P < 0.01 at P7, P14, and P60). (A3, A4) DRD2 and RGS9-2 levels show parallel courses, with a similar increase from P7 to P21 in $Tor1a^{+/+}$ as well as $Tor1a^{+/-}$ striatal lysates (P7: DRD2 N = 3, RGS9-2 N = 4; P14–P21 N = 4; one-way ANOVA with Bonferroni's multiple comparison test P > 0.05). At P60, a simultaneous reduction of DRD2 and RGS9-2 levels is observed in $Tor1a^{+/-}$ striatal lysates (N = 4; $Tor1a^{+/-}$ one-sample *t*-test DRD2: *P = 0.0250, RGS9-2: *P = 0.0175). WB quantification data, expressed as the ratio of protein vs. loading control intensity level, are normalized to the wild-type P60 sample of the same experiment. Mean ± SEM is represented in the graphs.

B    Representative WB images and the dot plot show downregulation of torsinA, DRD2, and RGS9-2 striatal levels also in adult P60–P90 $Tor1a^{\Delta gag/+}$ (Δgag/+) mice. TorsinA: $Tor1a^{+/+}$ N = 10, $Tor1a^{\Delta gag/+}$ N = 12, *t*-test ***P = 0.0006; DRD2: $Tor1a^{+/+}$ N = 7, $Tor1a^{\Delta gag/+}$ N = 9, *t*-test *P = 0.0183; RGS9-2: $Tor1a^{+/+}$ N = 10, $Tor1a^{\Delta gag/+}$ N = 9, *t*-test *P = 0.0192. WB quantification data, expressed as the ratio of protein vs. loading control intensity level, are normalized to the wild-type samples of the same experiment. Mean ± SEM is represented in the graph.

C    Left: Representative image of coronal striatal slices of fresh-frozen brains of wild-type and $Tor1a^{+/-}$ mice probed with the DRD2 radioligand ³H-spiperone. Scale bar 1.5 mm. Right: Graph showing DRD2 binding density in $Tor1a^{+/-}$ striatal sections, obtained from the densitometric quantification analysis, expressed as percentage variation compared to control animals. $Tor1a^{+/+}$ N = 5, $Tor1a^{+/-}$ N = 6, *t*-test **P = 0.0042. Mean ± SEM is represented.

Source data are available online for this figure.

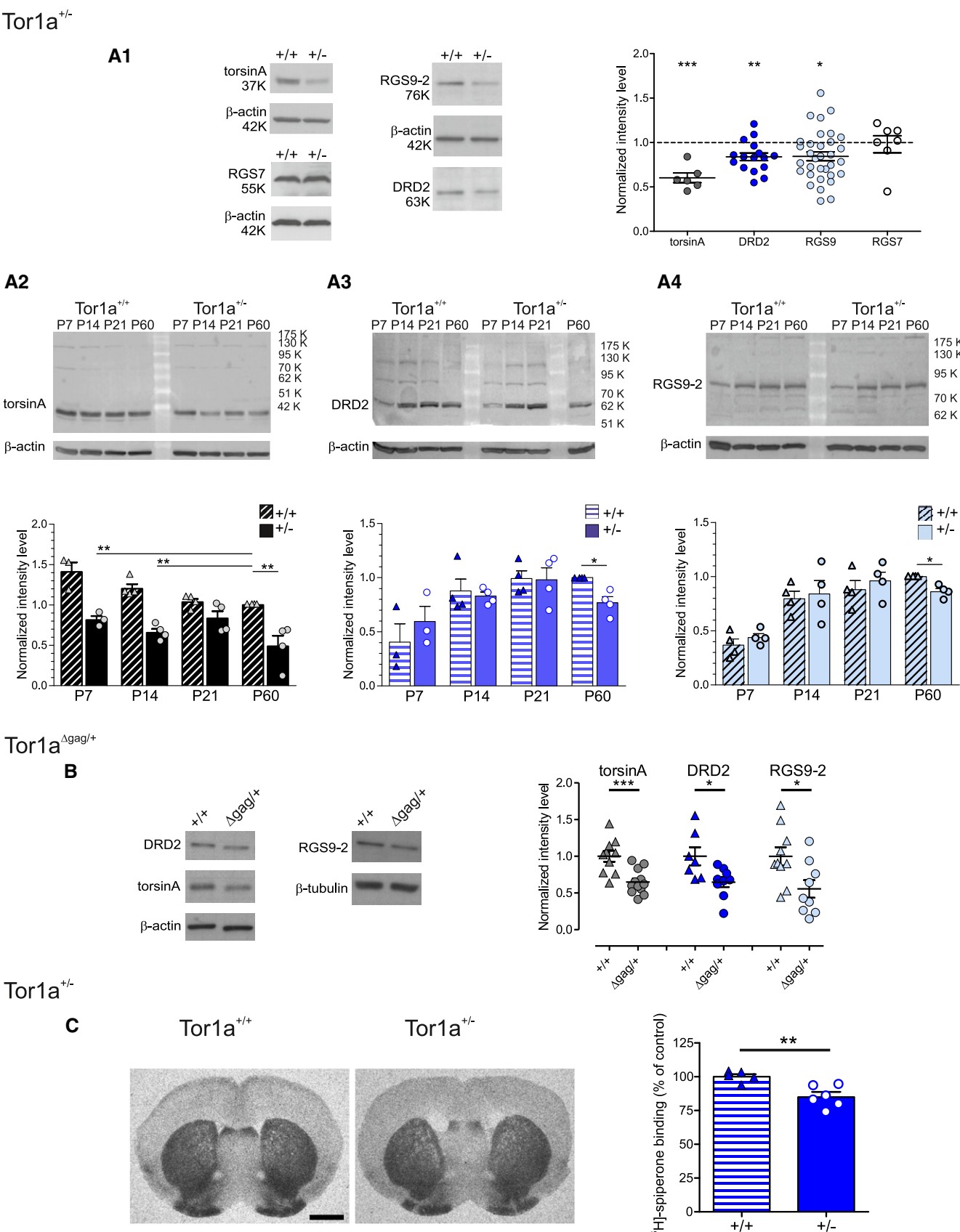

**Figure 1.**

2007). However, while the receptor remained at steady level at P60 in $Tor1a^{+/+}$ striatum, it showed a significant reduction in P60 $Tor1a^{+/-}$ samples (DRD2 P60 $Tor1a^{+/-}$ 0.768 ± 0.055, $N = 4$, one-sample $t$-test $P = 0.0250$). Interestingly, in $Tor1a^{+/+}$ mice the developmental profile of RGS9-2 protein expression (Anderson *et al*, 2007a) was superimposable to that of DRD2 (Pearson's $r$ correlation test: $R^2$ 0.97, $P = 0.0152$). Similarly, in $Tor1a^{+/-}$ striatum, RGS9-2 expression levels paralleled the course of DRD2 protein and showed a significant decrease at P60 (Fig 1$A_4$; RGS9-2 P60 $Tor1a^{+/-}$ 0.861 ± 0.029, $N = 4$, one-sample $t$-test $P = 0.0175$).

To confirm the involvement of DRD2 dysregulation in DYT1 dystonia, we measured striatal receptor levels in a different model, the $Tor1a^{\Delta gag/+}$ mouse. TorsinA levels were reduced in adult (P60–P90) male mutant mice with respect to wild-type littermates (Fig 1B; $Tor1a^{+/+}$ 1.000 ± 0.075 $N = 10$, $Tor1a^{\Delta gag/+}$ 0.649 ± 0.048 $N = 12$; $t$-test $P = 0.0006$), providing further evidence that the Δgag is a loss of function mutation that alters torsinA protein expression level (Goodchild *et al*, 2005; Gordon & Gonzalez-Alegre, 2008). Of note, DRD2 and RGS9-2 protein expression levels were also reduced in $Tor1a^{\Delta gag/+}$ striatum (Fig 1B; DRD2: $Tor1a^{+/+}$ 1.000 ± 0.123 $N = 7$, $Tor1a^{\Delta gag/+}$ 0.648 ± 0.067 $N = 9$, $t$-test $P = 0.0183$; RGS9-2: $Tor1a^{+/+}$ 1.000 ± 0.123 $N = 10$, $Tor1a^{\Delta gag/+}$ 0.556 ± 0.119 $N = 9$, $t$-test $P = 0.0192$), similarly to what we observed in $Tor1a^{+/-}$ mice, further strengthening the relevance of dopaminergic dysfunction in DYT1 dystonia.

Finally, we performed an *in situ* DRD2 binding study with ³H-spiperone (Zeng *et al*, 2004). As reported in Fig 1C, a significant 15% decrease in DRD2 binding density was found in Tor1a$^{+/-}$ as compared to wild-type striatum, in accordance with Western blot analysis (Fig 1C; Tor1a$^{+/+}$ 100.00 ± 1.83, $N = 5$; Tor1a$^{+/-}$ 84.94 ± 3.83, $N = 6$, $t$-test $P = 0.0042$).

## DRD2 downregulation in $Tor1a^{+/-}$ striatum is mediated by lysosomal degradation

RGS9-2 possesses specific determinants which target the protein for constitutive degradation by lysosomal proteases, unless it is shielded by its membrane anchor R7 binding protein (R7BP; Anderson *et al*, 2007b). Additionally, RGS9-2 interacts with another binding partner, the atypical G protein subunit type 5 G protein beta (Gβ5) subunit (Masuho *et al*, 2011). We therefore investigated whether RGS9-2 downregulation may be determined by changes in striatal levels of its binding partners. Unexpectedly, we found that striatal levels of R7BP were significantly increased, whereas Gβ5 protein amount was unaltered (Fig 2$A_1$–$A_3$; Gβ5 $Tor1a^{+/+}$: 1.000 ± 0.049 $N = 11$, $Tor1a^{+/-}$: 1.002 ± 0.036 $N = 10$, Mann–Whitney test $P = 0.2453$; R7BP $Tor1a^{+/+}$: 1.000 ± 0.057 $N = 12$, $Tor1a^{+/-}$: 1.279 ± 0.051 $N = 12$, $t$-test $P = 0.0014$). Therefore, we evaluated the stability of RGS9-2 protein, by measuring its degradation by lysosomal proteases. $Tor1a^{+/+}$ and $Tor1a^{+/-}$ dorsal striatum contralateral slices were incubated with or without the lysosomal protease inhibitor leupeptin (Anderson *et al*, 2007a; leu, 100 μM; Fig 2$B_1$ and $B_2$). In the absence of lysosomal proteolysis inhibition, we observed a similar extent of RGS9-2 degradation in $Tor1a^{+/-}$ and wild-type mice (Fig 2$B_2$; $Tor1a^{+/+}$: with leu 1.000 ± 0.115 $N = 5$, without leu 0.712 ± 0.041 $N = 5$, $t$-test $P = 0.0464$; $Tor1a^{+/-}$: with leu 0.917 ± 0.095 $N = 5$, without leu 0.610 ± 0.043 $N = 5$, $t$-test $P = 0.0190$; $Tor1a^{+/+}$ 74.00 ± 7.58% reduction; $Tor1a^{+/-}$ 70.58 ± 10.26% reduction, $Tor1a^{+/+}$ vs. $Tor1a^{+/-}$ $t$-test $P = 0.7956$), indicating that RGS9-2 protein turnover is unaltered in mutant mice. Since also the long-lived DRD2 protein can be targeted to lysosomal degradation (Li *et al*, 2012), we wondered whether the lower DRD2 levels in the $Tor1a^{+/-}$ striatum could be caused by increased lysosomal degradation. Thus, we quantified DRD2 protein expression level in the previously described experimental conditions, and found similar levels in lysates of $Tor1a^{+/+}$ striatal slices incubated with or without leu, indicating a long protein half-life (Fig 2$B_2$; $Tor1a^{+/+}$: with leu 1.000 ± 0.049 $N = 4$, without leu 0.925 ± 0.076 $N = 4$, $t$-test $P = 0.4376$). Surprisingly, however, in $Tor1a^{+/-}$ striatal lysates incubated without leu we observed a reduced level of the DRD2 protein (Fig 2$B_2$; $Tor1a^{+/-}$: with leu 0.904 ± 0.088 $N = 4$, without leu 0.664 ± 0.030 $N = 4$, $t$-test $P = 0.0427$), suggesting that loss of torsinA may disrupt DRD2 protein stability.

Detergent-resistant membranes (DRM) are thought to contain structures such as lipid rafts and to be involved in membrane compartmentalization. Notably, torsinA levels affect cellular lipid metabolism (Grillet *et al*, 2016) and therefore may alter the portion of DRD2 and RGS9-2 that is expressed in the DRM fraction. We therefore measured the level of DRD2 and RGS9-2 expressed in the DRM of control and $Tor1a^{+/-}$ striatum (Celver *et al*, 2012). We observed a reduction of DRD2 protein expression level in the DRM of $Tor1a^{+/-}$ striatum (Fig 2$C_1$; $Tor1a^{+/+}$ 1.000 ± 0.070 $N = 16$; $Tor1a^{+/-}$ 0.714 ± 0.083 $N = 13$; $t$-test $P = 0.0129$), further supporting a downregulation of the receptor. Notably, we found instead an increase in RGS9-2 in the DRM of $Tor1a^{+/-}$ striatum (Fig 2$C_2$; $Tor1a^{+/+}$ 1.000 ± 0.108 $N = 7$; $Tor1a^{+/-}$ 1.572 ± 0.234 $N = 6$; $t$-test $P = 0.0396$). This observation may explain the reduced RGS9-2 level in striatal lysates in the absence of changes in protein stability, suggesting an increased compartmentalization of the protein in the DRM. Indeed, the level of the RGS9-2 binding partner R7BP, a small SNARE-like membrane anchor which determines RGS plasma membrane localization (Drenan *et al*, 2006; Song *et al*, 2006; Anderson *et al*, 2007b), was increased in the $Tor1a^{+/-}$ striatum DRM with respect to control mice (Fig 2$C_3$; $Tor1a^{+/+}$ 1.000 ± 0.050 $N = 6$, $Tor1a^{+/-}$ 1.526 ± 0.079 $N = 6$; $t$-test $P = 0.0002$). Conversely, the level of Gβ5 subunit, which is constitutively associated with R7 RGS proteins (Masuho *et al*, 2011), was found unaltered in the DRM (Fig 2$C_4$; $Tor1a^{+/+}$ 1.000 ± 0.059 $N = 9$, $Tor1a^{+/-}$ 1.021 ± 0.089 $N = 9$; $t$-test $P = 0.8461$).

## DRD2 downregulation is mediated by endolysosomal trafficking and mimicked by RGS9-2 silencing

Our experiments provide evidence of normal DRD2 mRNA levels (Fig EV1), on one hand, but reduced protein stability and levels, on the other. We therefore investigated the pathway of protein quality control targeting plasma membrane proteins to endocytic trafficking and lysosomal degradation (MacGurn, 2014).

At membrane level, the reciprocal interactions of GPCRs with spinophilin and arrestin represent a regulatory mechanism for fine-tuning receptor-mediated signaling (Wang *et al*, 2004). β-Arrestin 2 (β-Arr2) is involved in internalization and G protein-independent signaling of DRD2 (Beaulieu *et al*, 2005), while spinophilin interacts with DRD2 and antagonizes arrestin actions (Smith *et al*, 1999;

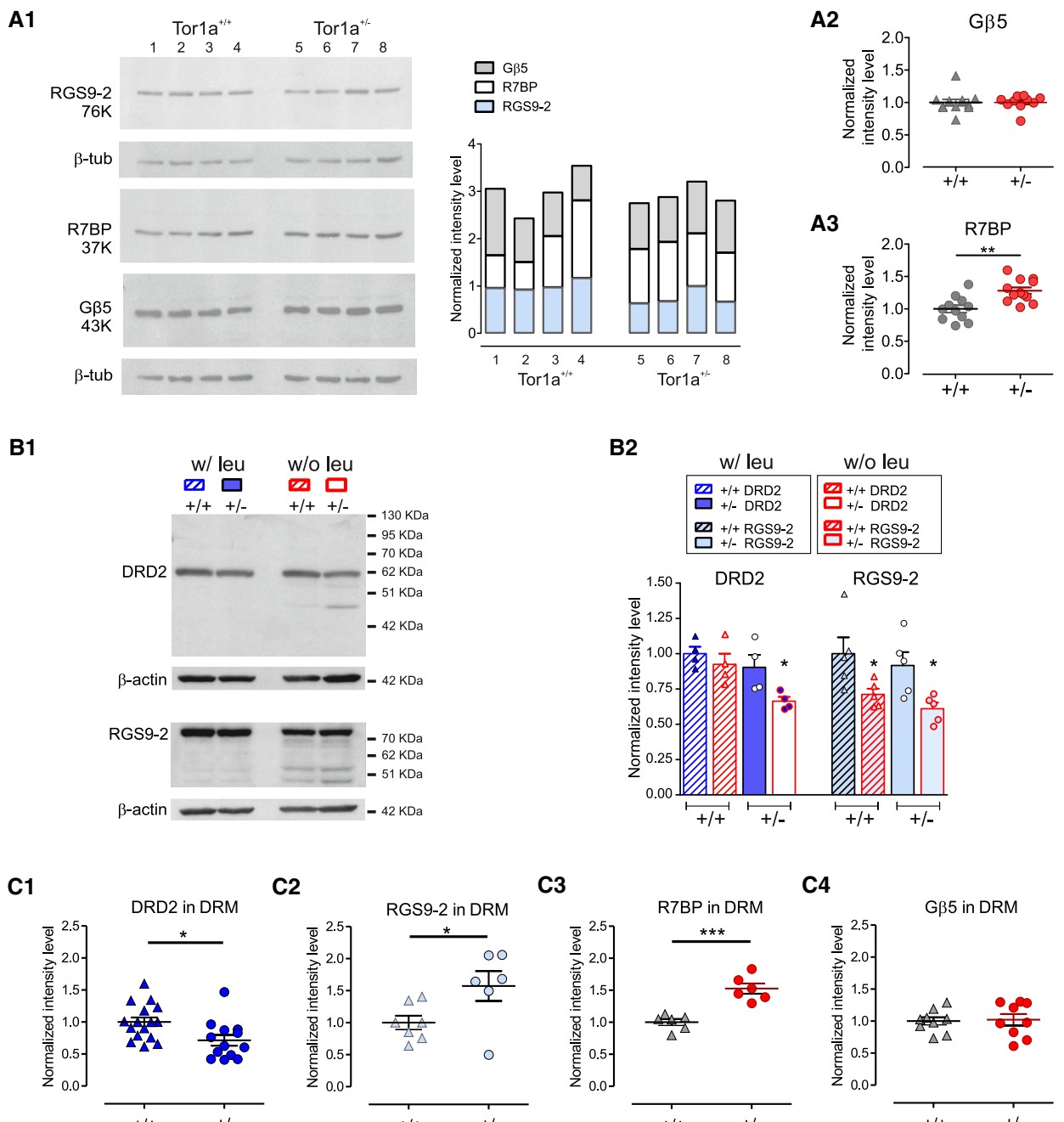

**Figure 2. DRD2 downregulation is mediated by lysosomal degradation.**

A  (A1) Representative WB of RGS9-2, R7BP, and Gβ5 proteins on four *Tor1a*$^{+/+}$ and four *Tor1a*$^{+/−}$ striatal lysates. The bar histogram on the right shows the quantification of the three proteins in each sample. (A2) Dot plot showing that Gβ5 striatal levels are unchanged in *Tor1a*$^{+/−}$ mice (*Tor1a*$^{+/+}$ N = 11, *Tor1a*$^{+/−}$ N = 10, Mann–Whitney test P = 0.2453). (A3) Conversely, the level of the R7 binding protein R7BP is increased in *Tor1a*$^{+/−}$ striatum (*Tor1a*$^{+/+}$ N = 12, *Tor1a*$^{+/−}$ N = 12, t-test **P = 0.0014). Values are represented as ratio of protein vs. loading control intensity level, normalized to the mean of *Tor1a*$^{+/+}$ control values of the same experiment. Mean ± SEM is represented.

B  (B1) Representative WB of lysates of *Tor1a*$^{+/+}$ and *Tor1a*$^{+/−}$ dorsal striatum slices incubated for 5 h in the presence (w/ leu) or absence (w/o leu; contralateral striatum) of the protease inhibitor leupeptin. Samples showing enhanced degradation of DRD2 and/or RGS9-2 proteins show additional bands at lower molecular weight. (B2) Summary plot reporting mean ± SEM of RGS9-2 and DRD2 protein level values, expressed as the ratio of protein vs. loading control intensity level, normalized to the value of the *Tor1a*$^{+/+}$ w/leu sample measured in the same experiment. DRD2: *Tor1a*$^{+/+}$ N = 4, t-test P = 0.4376; *Tor1a*$^{+/−}$ N = 4, t-test *P = 0.0427; RGS9-2: *Tor1a*$^{+/+}$ N = 5, t-test *P = 0.0464; *Tor1a*$^{+/−}$ N = 5, t-test *P = 0.0190.

C  Dot plots showing DRD2, RGS9-2, R7BP, and Gβ5 protein levels measured in striatal detergent-resistant-membrane (DRM) preparations from *Tor1a*$^{+/−}$ (+/−) and WT (+/+) mice (DRD2: *Tor1a*$^{+/+}$ N = 16, *Tor1a*$^{+/−}$ N = 13, t-test *P = 0.0129; RGS9-2: *Tor1a*$^{+/+}$ N = 7, *Tor1a*$^{+/−}$ N = 6, t-test *P = 0.0396; R7BP: *Tor1a*$^{+/+}$ N = 6, *Tor1a*$^{+/−}$ N = 6, t-test ***P = 0.0002; Gβ5: *Tor1a*$^{+/+}$ N = 9, *Tor1a*$^{+/−}$ N = 9, t-test P = 0.8461). Values are reported as ratio of protein vs. PSD-95 intensity level, normalized to the mean value of the *Tor1a*$^{+/+}$ samples measured in the same experiment. Mean ± SEM is represented.

Source data are available online for this figure.

Wang *et al*, 2004). WB quantification of β-Arr2 showed no significant differences in lysates from *Tor1a*$^{+/+}$ and *Tor1a*$^{+/-}$ striata (Fig 3A; *Tor1a*$^{+/+}$ 1.033 ± 0.059 N = 11; *Tor1a*$^{+/-}$ 1.088 ± 0.119 N = 12; *t*-test P = 0.6796). Conversely, the level of spinophilin was significantly reduced in *Tor1a*$^{+/-}$ striatum (Fig 3B; *Tor1a*$^{+/+}$ 1.000 ± 0.010 N = 5; *Tor1a*$^{+/-}$ 0.597 ± 0.091 N = 8; *t*-test P = 0.0151), suggesting an imbalance in β-Arr2-mediated actions that may favor DRD2 removal from the plasma membrane. Of note, β-Arr2 co-immunoprecipitates with RGS9-2 (Zheng *et al*, 2011) and RGS9-2 inhibits DRD2 internalization (Celver *et al*, 2010). Thus, we asked whether changes in RGS9-2 content would affect DRD2 level. To address this issue, we silenced RGS9-2 in wild-type mouse dorsal striatum. ShRNA-mediated downregulation of RGS9-2 caused a

decrease in striatal DRD2 protein level with respect to the sham contralateral striatum (Fig 3C; RGS9-2: sham 0.598 ± 0.063, sh-RGS9-2 0.308 ± 0.026, N = 4, paired *t*-test P = 0.0100; DRD2: sham 0.622 ± 0.046, sh-RGS9-2 0.398 ± 0.059, N = 4; paired *t*-test P = 0.0432), suggesting that RGS9-2-DRD2 interaction prevents β-Arr2-mediated constitutive recycling and downregulation of the receptor. Indeed, DRD2 constitutive recycling through Rab4-expressing endosomes serves as a quality control system, by sorting receptors toward either the plasma membrane or degradation pathways (Li *et al*, 2012). In line with our hypothesis, we found an increased level of Rab4 in *Tor1a*$^{+/-}$ striatal samples (Fig 3D *Tor1a*$^{+/+}$ 1.000 ± 0.058 N = 13; *Tor1a*$^{+/-}$ 1.301 ± 0.087 N = 14; *t*-test P = 0.0087).

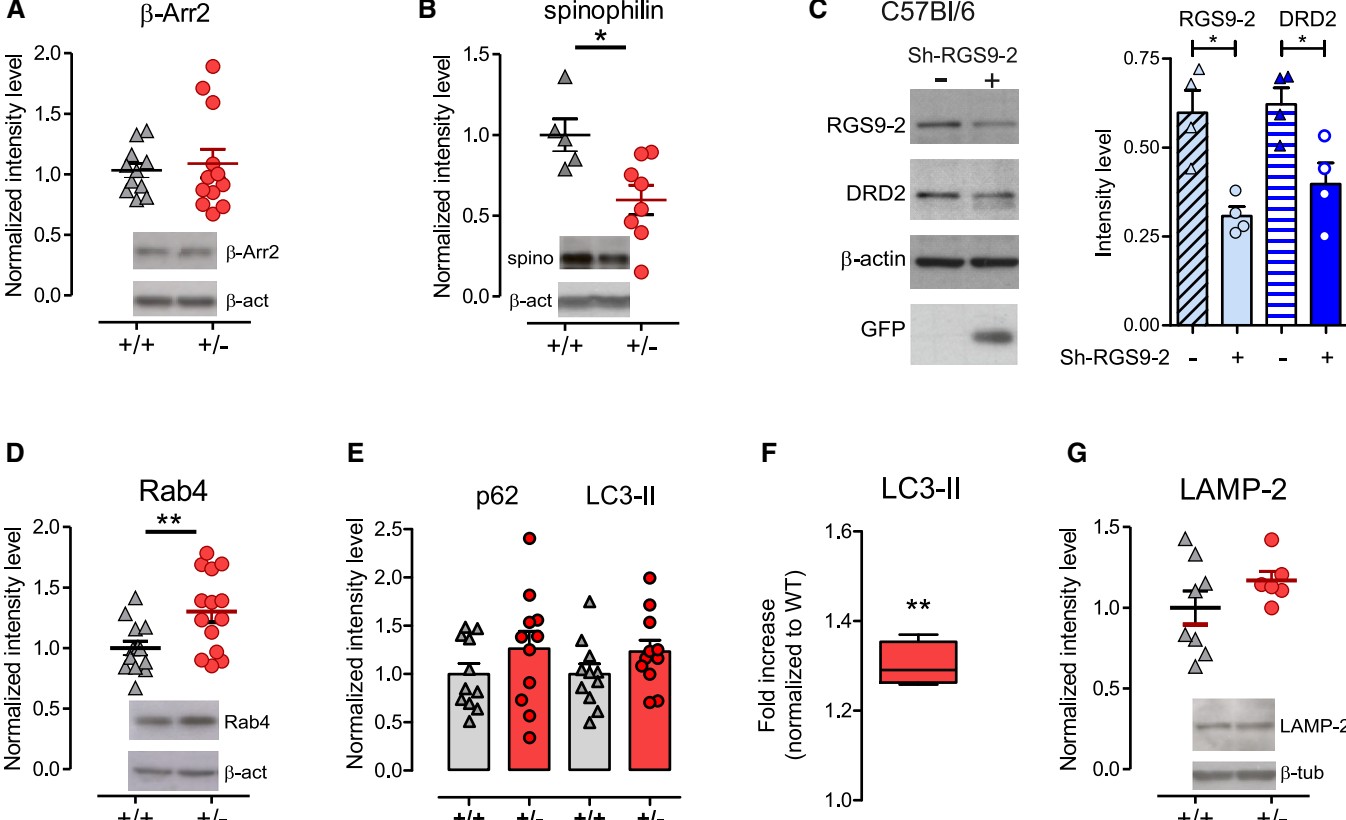

**Figure 3.    DRD2 downregulation is mediated by enhanced degradative trafficking and mimicked by RGS9-2 silencing.**

WB analysis of markers of endocytic trafficking and degradative pathway in the dorsal striatum of *Tor1a*$^{+/-}$ mice.

A, B    (A) The level of β-arrestin 2 (β-Arr2) is not altered (*Tor1a*$^{+/+}$ N = 11, *Tor1a*$^{+/-}$ N = 12, *t*-test P = 0.6796), whereas the level of spinophilin (B) is decreased (*Tor1a*$^{+/+}$ N = 5, *Tor1a*$^{+/-}$ N = 8, *t*-test *P = 0.0151).

C    A Sh-RGS9-2-GFP viral preparation injected into the right striatum (+) of C57BL/6 mice effectively reduces both RGS9-2 and DRD2 protein levels, with respect to the contralateral striatum (−).The summary plot reports mean ± SEM values (RGS9-2: N = 4, paired *t*-test *P = 0.0100; DRD2: N = 4, paired *t*-test *P = 0.0432).

D    The endosomal marker Rab4 is upregulated in the *Tor1a*$^{+/-}$ striatum (*Tor1a*$^{+/+}$ N = 13, *Tor1a*$^{+/-}$ N = 14, *t*-test **P = 0.0087).

E    Basal levels of markers of the degradative pathway, p62 and LC3-II, show a trend toward upregulation in *Tor1a*$^{+/-}$ striatal lysates (p62: *Tor1a*$^{+/+}$ N = 11, *Tor1a*$^{+/-}$ N = 11, *t*-test P = 0.2292; LC3-II: *Tor1a*$^{+/+}$ N = 9, *Tor1a*$^{+/-}$ N = 11, *t*-test P = 0.1574).

F    Treatment with bafilomycin A$_1$ (BafA$_1$) induces a significantly more pronounced increase in LC3-II in *Tor1a*$^{+/-}$ than in wild-type dorsal striatum slices. BafA1-induced increase in LC3-II level measured in *Tor1a*$^{+/-}$ slices was normalized to the increase observed in the *Tor1a*$^{+/+}$ samples of the same experiment (N = 4, one-sample *t*-test **P = 0.0012). The bottom and top edges of the box indicate the 25[th] and 75[th] percentiles, respectively; the line indicates the median value; and the whiskers indicate the minimum and maximum values.

G    LAMP-2, a lysosomal marker, shows a trend for upregulation (*Tor1a*$^{+/+}$ N = 8; *Tor1a*$^{+/-}$ N = 6, *t*-test P = 0.2232).

Data information: The graphs in (A, B, D, E, G) report mean ± SEM of the ratio of protein vs. loading control intensity level, normalized to the *Tor1a*$^{+/+}$ controls of the same experiment.

Source data are available online for this figure.

The autophagy–lysosomal pathway operates constitutively at low rate and exerts a key role in quality control and turnover of long-lived proteins (Hara *et al*, 2006; Komatsu & Ichimura, 2010). The ubiquitin- and LC3-binding protein p62 (also known as SQSTM1) links targeted proteins to the microtubule-associated protein light chain 3 (LC3), an ubiquitin-like modifier that, in its lipidated form LC3-II, plays crucial roles in the formation of autophagosomes (Bjørkøy *et al*, 2005; Tanida *et al*, 2005). We thus measured striatal levels of p62 and LC3-II to monitor this pathway, and observed a nearly 25% increase, though not statistically significant, of both proteins in $Tor1a^{+/-}$ mice (Fig 3E; p62: $Tor1a^{+/+}$ 1.000 ± 0.111 $N = 11$, $Tor1a^{+/-}$ 1.263 ± 0.180 $N = 11$, *t*-test $P = 0.2292$; LC3-II: $Tor1a^{+/-}$ 1.100 ± 0.100 $N = 9$, $Tor1a^{+/-}$ 1.233 ± 0.118 $N = 11$, *t*-test $P = 0.1574$). We therefore utilized bafilomycin $A_1$, a blocker of the autophagosome–lysosome fusion and acidification, in order to prevent LC3-II degradation into autophagolysosomes. In this experimental condition, a more pronounced increase in LC3-II level is observed in $Tor1a^{+/-}$ with respect to wild-type striatum (Fig 3F; $Tor1a^{+/-}$ 1.303 ± 0.025 $N = 4$; one-sample *t*-test $P = 0.0012$), indicating an increased lysosomal turnover of the autophagosomal marker LC3-II (Tanida *et al*, 2008). Then, to measure the number of lysosomes in $Tor1a^{+/-}$ striatum, we used the marker lysosome-associated membrane protein-2 (LAMP-2; Eskelinen, 2006). WB of dorsal striatum lysates showed a trend for an increase in LAMP-2 levels in mutant mice (Fig 3G; $Tor1a^{+/+}$ 1.000 ± 0.104 $N = 8$; $Tor1a^{+/-}$ 1.168 ± 0.058 $N = 6$; *t*-test $P = 0.2232$). Overall, these data indicate a selective enhancement of the endosomal trafficking and lysosomal-mediated degradation of DRD2 in the $Tor1a^{+/-}$ striatum.

## The autophagy–lysosomal pathway is upregulated in striatal DRD2-expressing cholinergic interneurons

Our previous work consistently demonstrated electrophysiological alterations of DRD2 in the population of striatal cholinergic interneurons (ChIs) in multiple *DYT1* dystonia models (Pisani *et al*, 2006; Sciamanna *et al*, 2011, 2012a; Martella *et al*, 2014). We therefore wondered whether increased receptor turnover could be responsible for the observed DRD2 dysfunction in *DYT1* ChIs. Indeed, we observed a significant enhancement of p62 signal in ChIs, identified by choline acetyltransferase (ChAT) immunolabeling in $Tor1a^{+/-}$ striatal sections (Fig 4$A_1$–$A_3$; $Tor1a^{+/+}$ 1.000 ± 0.086 $n = 20$ cells, $N = 6$ mice; $Tor1a^{+/-}$ 1.648 ± 0.120 $n = 20$ cells, $N = 6$ mice; *t*-test $P = 0.00008$).

Next, in order to measure autophagy flux *in vivo*, specifically in ChIs, we utilized as a reporter an adeno-associated virus carrying a monomeric tandem mCherry-GFP-LC3 construct (AAV2/9-mCherry-GFP-LC3; Castillo *et al*, 2013). The GFP fluorescent signal of the reporter is sensitive to acidic conditions; thus, co-localization of green and red fluorescence (yellow puncta) indicates that the tandem protein is not localized in compartments fused with a lysosome, while detection of red puncta indicates that the protein is located in the autophagolysosome (Matus *et al*, 2014). Figure 5A shows representative images of successful viral transduction of the dorsal striatum of control and $Tor1a^{+/-}$ mice, 4 weeks after stereotactic delivery of AAV2/9-mCherry-GFP-LC3. Quantification of the mCherry/ChAT co-localization demonstrated a significant increase in mCherry signal in ChIs from $Tor1a^{+/-}$ mice (Fig 5B; $Tor1a^{+/+}$

1.00 ± 0.09 $n = 24$ cells, $N = 8$ mice; $Tor1a^{+/-}$ 1.69 ± 0.15 $n = 22$ cells, $N = 8$ mice; *t*-test $P = 0.0005$). The ratio of mCherry/GFP fluorescence per cell area, measuring LC3 flux, was unchanged (Fig 5B; $Tor1a^{+/+}$ 16.85 ± 2.89 $n = 19$ cells; $Tor1a^{+/-}$ 18.27 ± 2.37 $n = 21$ cells; Mann–Whitney test $P = 0.2334$). Then, we quantified red and yellow dots in ChAT-positive neurons and found a significant increase in both mCherry dots (Fig 5C and D; $Tor1a^{+/+}$ 26.38 ± 1.95 dots/cell $n = 13$ cells, $Tor1a^{+/-}$ 46.21 ± 3.98 dots/cell $n = 14$ cells; $N = 5$ mice/genotype; *t*-test $P = 0.0003$) and yellow puncta (Fig 5B and C; $Tor1a^{+/+}$ 3.62 ± 0.43 dots/cell, $n = 13$ cells, $Tor1a^{+/-}$ 7.14 ± 0.99 dots/cell, $n = 14$ cells; $N = 5$ mice/genotype; *t*-test $P = 0.0048$) in ChIs from $Tor1a^{+/-}$ mice, indicating an increase in LC3 in all the compartments of the autophagy–lysosomal pathway. Indeed, measurement of the mCherry/yellow ratio indicated that the autophagic flux was not altered (Fig 5D; $Tor1a^{+/+}$ 9.65 ± 2.16 dots/cell, $n = 13$ cells; $Tor1a^{+/-}$ 9.14 ± 1.75 dots/cell, $n = 14$ cells; $N = 5$ mice/genotype; Mann–Whitney test $P = 0.8842$).

Immunoblotting and immunohistochemical data jointly suggest an increased trafficking of DRD2 from the plasma membrane to the endolysosomal pathway, causing its downregulation. Nevertheless, reduced torsinA levels in *DYT1* mice may cause DRD2 retention into the endoplasmic reticulum (ER), where torsinA resides (Cascalho *et al*, 2017) and may influence receptor protein maturation and export processes. However, our confocal microscopy analysis (Fig 6A, $B_1$ and $B_2$) seems to rule out this possibility, as we did not observe co-localization of DRD2 with the ER marker protein disulfide isomerase (PDI; Fig 6$B_1$ and $B_2$).

## Restoration of RGS9-2 protein levels rescues DRD2 level and function

Our gene interference experiments showed that viral-mediated downregulation of RGS9-2 causes a reduction in striatal DRD2 protein level in wild-type mice. We therefore asked whether, by increasing RGS9-2 protein amount, we could rescue DRD2 level in the *DYT1* mouse striatum. We thus injected into the dorsal striatum of $Tor1a^{+/-}$ mice either a herpesvirus (HSV-RGS9-2; Gold *et al*, 2007) or a lentivirus (LV-RGS9-2) both carrying the RGS9-2 construct, while the contralateral striatum was injected with the respective sham vectors (HSV-LacZ or LV-GFP, respectively; Fig 7A). The two different vectors were chosen because the HSV-mediated transduction is transient and RGS9-2 shows a peak of expression at 3–5 days after injection, whereas the LV-induced maximal expression is reached more slowly (3 weeks) but is stable over time (Mandolesi *et al*, 2009; Sciamanna *et al*, 2012b). Both HSV-RGS9-2 and LV-RGS9-2 delivery rescued RGS9-2 content in $Tor1a^{+/-}$ striatum to control level (Fig 7B and C; $Tor1a^{+/+}$: 1.000 ± 0.044 $N = 10$; $Tor1a^{+/-}$: HSV/LV-RGS9-2-injected striatum 1.038 ± 0.033 $N = 7$; $Tor1a^{+/-}$ contralateral striatum 0.696 ± 0.037 $N = 7$; $Tor1a^{+/-}$ injected vs. contralateral: paired *t*-test $P = 0.0011$). Notably, $Tor1a^{+/-}$ striata showing a restoration of RGS9-2 level (Fig 7B, lanes 4 and 5), indicating a successful infection, also showed a complete rescue of DRD2 level (Fig 7B and D; $Tor1a^{+/+}$: 1.000 ± 0.038 $N = 9$, $Tor1a^{+/-}$: HSV/LV-RGS9-2-injected 1.124 ± 0.153 $N = 6$; $Tor1a^{+/-}$: contralateral striatum 0.671 ± 0.079 $N = 6$; $Tor1a^{+/-}$ injected vs. contralateral: paired *t*-test $P = 0.0380$). Finally, in order to obtain a functional rescue, we tested the electrophysiological responses to DRD2 activation after

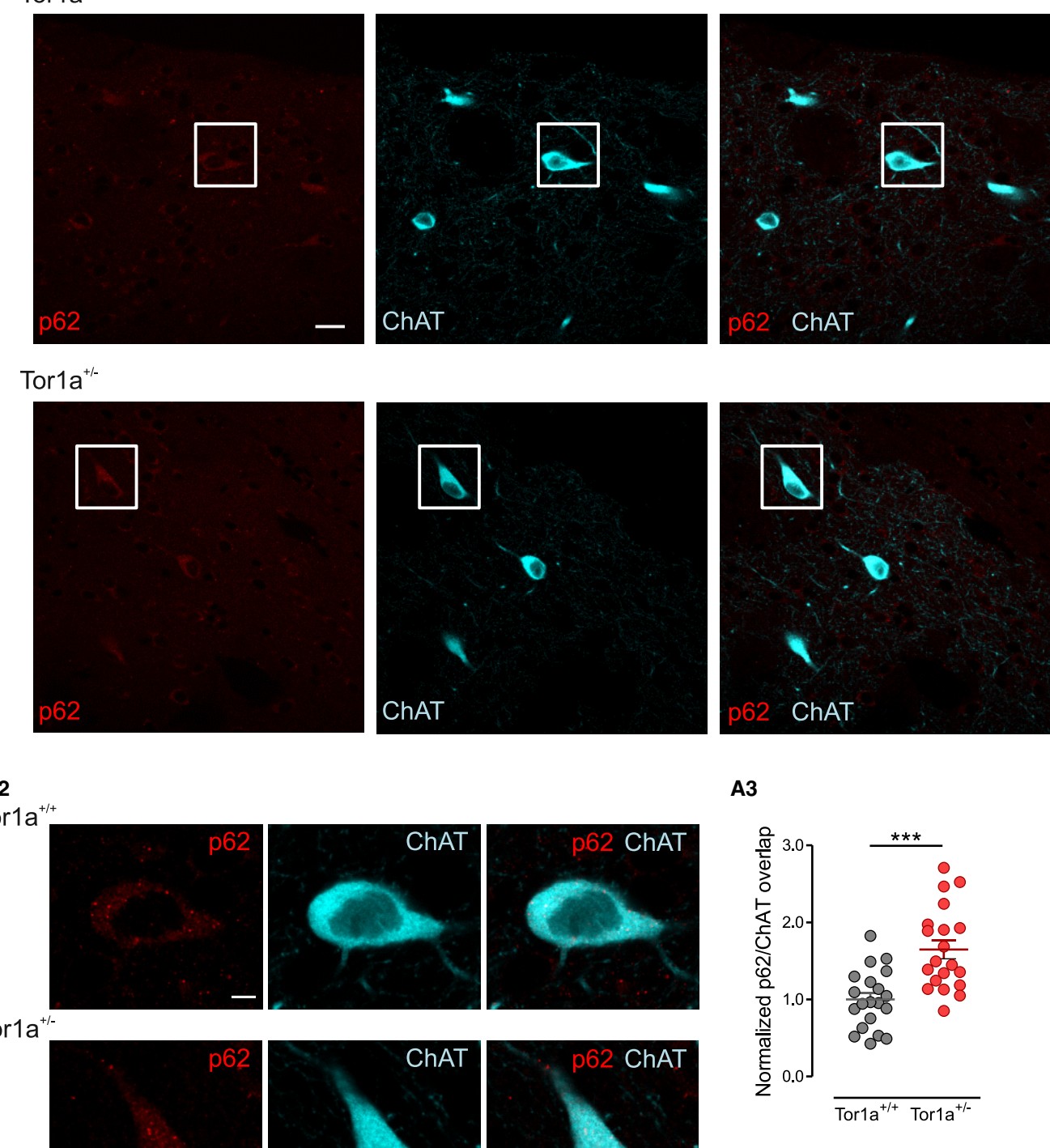

**Figure 4.  Increased p62 immunolabeling in striatal ChIs.**

Representative double-immunofluorescence confocal images of coronal sections of the dorsal striatum showing p62 immunolabeling (red) in choline acetyltransferase (ChAT)-positive (cyano) ChIs of *Tor1a*[+/+] and *Tor1a*[+/−] mice. (A1) Two representative ChIs identified by the white box in the low-magnification images (scale bar = 20 μm) are shown at higher magnification in (A2) (scale bar = 5 μm). (A3) The graph reporting mean ± SEM values of p62/ChAT signal overlap in ChIs, normalized to *Tor1a*[+/+] values of the same experiment, shows a significant increase in p62 in *Tor1a*[+/−] neurons (*Tor1a*[+/+] $n = 20$ cells, $N = 6$ mice; *Tor1a*[+/−] $n = 20$ cells, $N = 6$ mice, *t*-test ***$P = 0.00008$).

**Figure 5.  The AAV2/9-mCherry-GFP-LC3 reporter indicates upregulation of the autophagy pathway in ChIs.**

A    Top: Low-magnification merged confocal images of coronal striatal sections showing ChAT-labeled (cyano) ChIs in the area of mCherry fluorescence (red) indicating viral infection. Scale bar = 100 μm. Bottom: Higher magnification images of the ChIs identified by the white box in the upper images show viral-induced expression of mCherry and GFP signal. Scale bar = 20 μm.

B    The mCherry/ChAT overlap is increased in ChIs from $Tor1a^{+/-}$ mice ($Tor1a^{+/+}$ $n$ = 24 cells, $Tor1a^{+/-}$ $n$ = 22 cells, $N$ = 8 mice/genotype, $t$-test ***$P$ = 0.0005), but the mCherry/GFP fluorescence ratio indicated that the LC3 flux was unchanged ($Tor1a^{+/+}$ $n$ = 19 cells, $Tor1a^{+/-}$ $n$ = 21 cells, $N$ = 8 mice/genotype, Mann–Whitney test $P$ = 0.2334).

C, D    (C) Merged and split channel confocal images (scale bar = 10 μm) of representative $Tor1a^{+/+}$ and $Tor1a^{+/-}$ ChIs showing red dots and yellow puncta (indicated by white circles), quantified in (D): red dots: $Tor1a^{+/+}$ $n$ = 13 cells, $Tor1a^{+/-}$ $n$ = 14 cells, $t$-test ***$P$ = 0.0003; yellow dots: $Tor1a^{+/+}$ $n$ = 13 cells, $Tor1a^{+/-}$ $n$ = 14 cells, $t$-test **$P$ = 0.0048, $N$ = 5 mice/genotype. Measurement of the mCherry/yellow dots ratio indicated that the autophagic flux was not altered ($Tor1a^{+/+}$ $n$ = 13 cells; $Tor1a^{+/-}$ $n$ = 14 cells, Mann–Whitney test $P$ = 0.8842).

Data information: In (B and D) mean ± SEM is represented.
Source data are available online for this figure.

restoration of RGS9-2 content. Indeed, though most *DYT1* rodent models do not express an overt motor phenotype, they however share a common electrophysiological alteration consisting in an aberrant excitatory response of striatal ChIs to DRD2 activation, instead of the physiological inhibition recorded in wild-type littermates (Pisani *et al*, 2006; Sciamanna *et al*, 2011, 2012a; Martella *et al*, 2014). We therefore performed patch-clamp recordings from ChIs of different DYT1 mouse models in the area of viral transduction, as confirmed by post-recording immunohistochemistry (Fig 8A), either 3–5 days or 3 weeks after injection of HSV-RGS9-2 or LV-RGS9-2, respectively. Initially, we analyzed the effect of HSV/LV-RGS9-2 in two DYT1 mouse models where we had consistently reported an abnormal excitatory response of striatal ChIs to the DRD2 agonist quinpirole (10 μM, 2 min), transgenic hMT mice ($N$ = 6), and knock-in $Tor1a^{\Delta gag/+}$ ($N$ = 3; Pisani *et al*, 2006; Sciamanna *et al*, 2011; Martella *et al*, 2014). As shown in Fig 8A,

HSV/LV-RGS9-2 infection restored a physiological effect of quinpirole, consisting in a reduction of action potential firing frequency, with respect to the contralateral striatum, both in $Tor1a^{\Delta gag/+}$ mice (LV-GFP 199.0 ± 75.88% of basal frequency, $n$ = 7; LV-RGS9-2 112.5 ± 13.52% of basal frequency, $n$ = 4; $t$-test $P$ = 0.3048) and in transgenic hMT mice (Fig 8B; HSV-LacZ 165.60 ± 18.25% of basal frequency, $n$ = 14; HSV-RGS9-2 39.49 ± 13.10% of basal frequency, $n$ = 9; $t$-test $P$ < 0.0001).

We now show that an abnormal excitatory response to DRD2 activation is observed also in ChIs from $Tor1a^{+/-}$ mice (Fig 8C; $Tor1a^{+/+}$: basal 1.589 ± 0.2901 Hz, in quinpirole 1.232 ± 0.2023 Hz, $N$ = 6, $n$ = 14, paired $t$-test $P$ = 0.0395; $Tor1a^{+/-}$: 1.739 ± 0.1801 Hz, in quinpirole 2.702 ± 0.2567, $N$ = 10, $n$ = 27, paired $t$-test $P$ = 0.0009). Expectedly, ChIs recorded from the sham-injected contralateral striatum of $Tor1a^{+/-}$ mice still showed an aberrant excitatory response to quinpirole. Conversely, neurons recorded from slices of HSV- and LV-

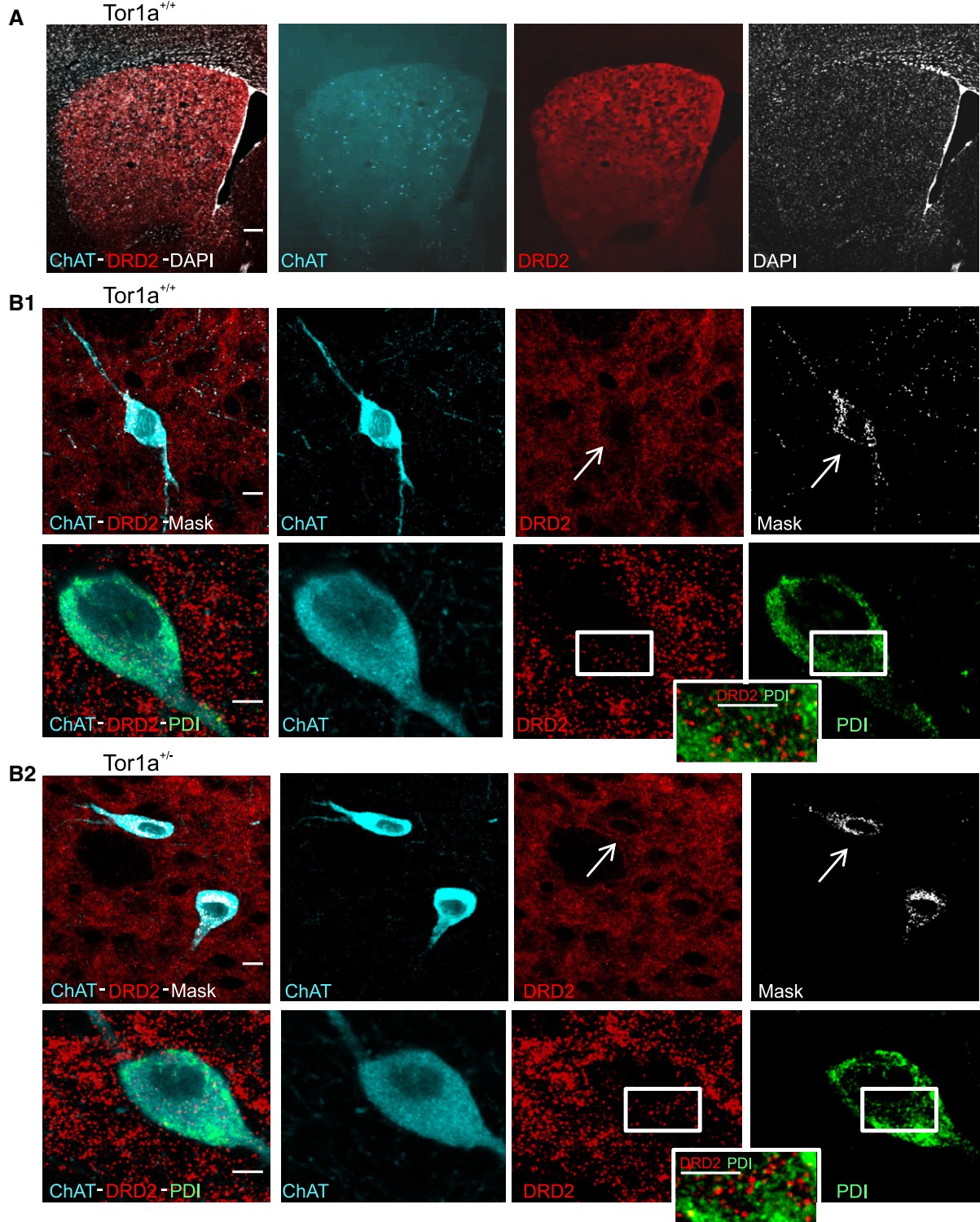

**Figure 6.  Subcellular localization of DRD2 in striatal ChIs.**

A   Low-magnification merged and split channel confocal images of striatal sections showing immunolabeling for ChAT (cyano), DRD2 (red), and the nuclear stain DAPI (white). Scale bar = 200 μm.

B   Top: Representative ChAT-positive (cyano) cholinergic interneurons, labeled for DRD2 (red), from a *Tor1a*$^{+/+}$ (B1) or *Tor1a*$^{+/−}$ (B2) striatal section. The co-localization mask of the two signals is shown in white (right). Arrows indicate the interneuron identified by ChAT immunolabeling. Scale bar = 10 μm. Bottom: Higher magnification images (scale bar = 5 μm) of representative *Tor1a*$^{+/+}$ (B1) and *Tor1a*$^{+/−}$ (B2) ChIs showing the lack of co-localization of DRD2 with the ER marker PDI (green). A detail at higher magnification is shown in the insets.

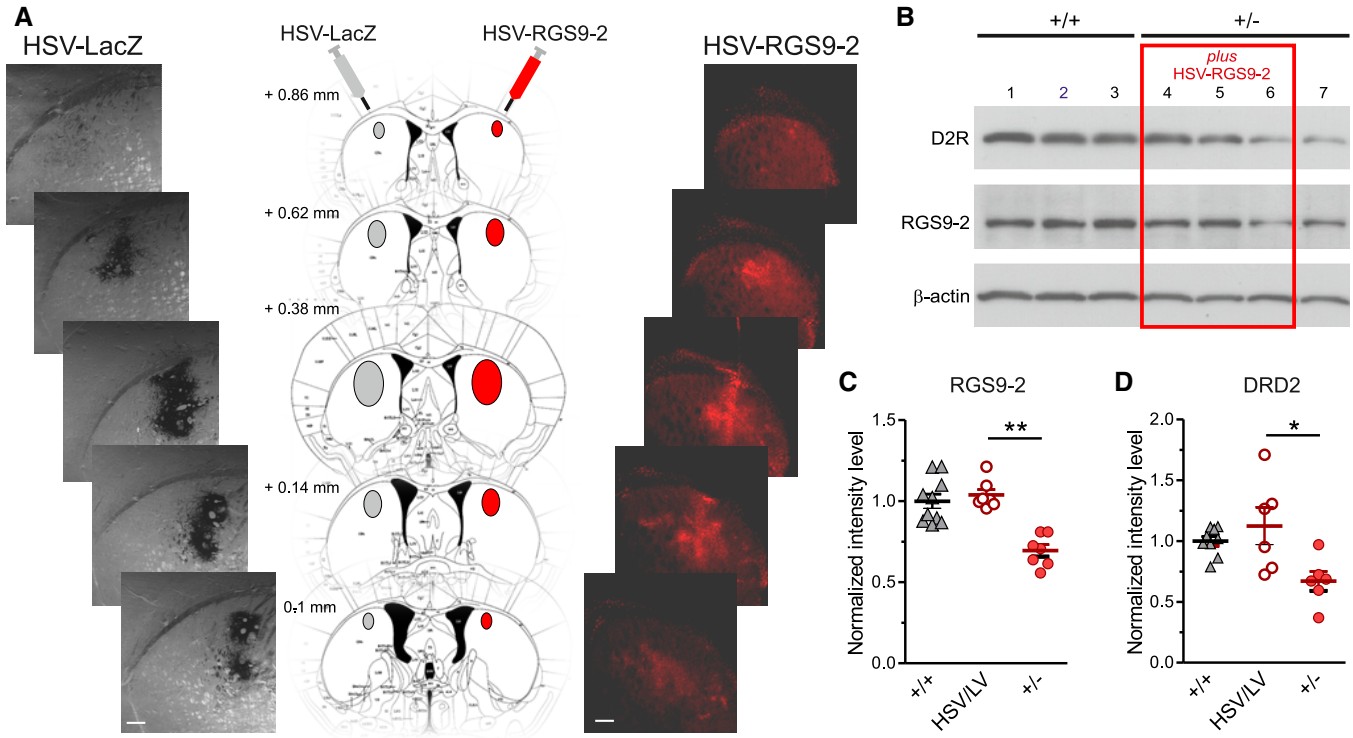

**Figure 7. *In vivo* viral-mediated delivery of RGS9-2 into *Tor1a*^{+/−} dorsal striatum rescues DRD2 protein level.**

A   Middle: Graphic reconstruction of an anteroposterior sequence of coronal sections of mouse striatum, reporting the coordinates relative to Bregma. HSV-LacZ (left, gray) and HSV-RGS9-2 (right, red) were injected into the left and right striata, respectively, and mice sacrificed after 3–5 days. Left: Serial confocal images showing the extent of viral infection, as indicated by the robust dark β-gal staining. Right: The RGS9-2 immunolabeling (red) is particularly intense near the injection site, indicating enhanced protein expression with respect to the endogenous level. Scale bar = 200 μm.

B   A representative WB shows correlated viral-mediated increases in RGS9-2 and DRD2 protein levels in *Tor1a*^{+/−} dorsal striata (lanes 4 and 5), while in a sample where RGS9-2 level was not increased by viral infection (lane 6), DRD2 level remained low as in the sham-injected striatum (lane 7).

C, D   The summary plots report mean ± SEM values of the ratio of protein vs. loading control intensity level measured from the injected (HSV/LV) or sham contralateral *Tor1a*^{+/−} (+/−) striata, normalized to the *Tor1a*^{+/+} (+/+) samples of the same experiment. (C) RGS9-2: *Tor1a*^{+/+} ($N = 10$); *Tor1a*^{+/−} injected ($N = 7$) vs. contralateral ($N = 7$): paired *t*-test **$P = 0.0011$; (D) DRD2: *Tor1a*^{+/+} ($N = 9$); *Tor1a*^{+/−} injected ($N = 6$) vs. contralateral ($N = 6$): paired *t*-test *$P = 0.0380$.

Source data are available online for this figure.

RGS9-2-infected *Tor1a*^{+/−} ($N = 6$) striata showed a recovery of the physiological inhibitory response to DRD2 activation (Fig 8D; LacZ/GFP: 239.30 ± 47.33% of basal frequency, $n = 8$; RGS9-2: 90.19 ± 27.75% of basal frequency, $n = 9$; *t*-test $P = 0.0136$). In ChIs from *Tor1a*^{+/+} mice ($N = 3$), we did not observe significant effects of

RGS9-2 overexpression on DRD2 function (Fig 8D; 79.08 ± 22.40% of basal frequency, $n = 7$).

In conclusion, viral vector-mediated restoration of striatal RGS9–2 content is effective in rescuing DRD2 protein expression and function in different mouse models of *DYT1* dystonia.

**Figure 8. RGS9-2 rescues DRD2 function in striatal ChIs.**

A   As depicted in the cartoon and shown by representative confocal images, patch-clamp recordings were performed in the dorsal striatum, in the area of HSV-LacZ (blue, left) and HSV-RGS9-2 (red, right) viral infection, the latter showing an increased RGS9-2 immunolabeling with respect to the sham contralateral striatum of the same mouse. Representative confocal images at higher magnification show co-localization of RGS9-2 immunolabeling with Lucifer Yellow (LY) fluorescence in two ChIs loaded intracellularly with LY during whole-cell recordings in the infected area. Scale bar = 10 μm.

B   Representative patch-clamp recordings of ChIs from hMT (top, perforated configuration) and *Tor1a*^{Δgag/+} (bottom, cell-attached configuration) mice, showing that viral-mediated delivery of RGS9-2 rescues the physiological response to the DRD2 agonist quinpirole (10 μM, 2 min). The dot plot reports the firing frequency changes induced by quinpirole in hMT striata ($N = 6$, HSV-RGS9-2 $n = 9$, HSV-LacZ $n = 14$, *t*-test ***$P < 0.0001$).

C   Representative traces show a typical response of *Tor1a*^{+/+} ChIs to quinpirole, characterized by a reduction in the frequency of the ongoing firing activity of the cell, as opposed to an aberrant increase in firing frequency in *Tor1a*^{+/−} ChIs, quantified in the plot (right, *Tor1a*^{+/+} $N = 6$, $n = 14$; *Tor1a*^{+/−} $N = 10$, $n = 27$; *Tor1a*^{+/+} vs. *Tor1a*^{+/−} in quinpirole *t*-test ***$P = 0.0005$; *Tor1a*^{+/−} basal vs. quinpirole paired *t*-test ***$P = 0.0009$).

D   LV-mediated delivery of RGS9-2 rescues the physiological response to quinpirole in *Tor1a*^{+/−} slices. Dot plot reporting firing frequency changes induced by quinpirole in *Tor1a*^{+/+} ($N = 3$) and *Tor1a*^{+/−} ($N = 6$) striatal ChIs infected with LV/HSV-RGS9-2 or control LV-GFP and HSV-LacZ ChIs (*Tor1a*^{+/+} RGS9-2 $n = 7$; *Tor1a*^{+/−} RGS9-2 $n = 9$; *Tor1a*^{+/−} LacZ/GFP $n = 8$; *Tor1a*^{+/−}: LacZ/GFP vs. RGS9-2 *t*-test *$P = 0.0136$). Mean ± SEM is represented.

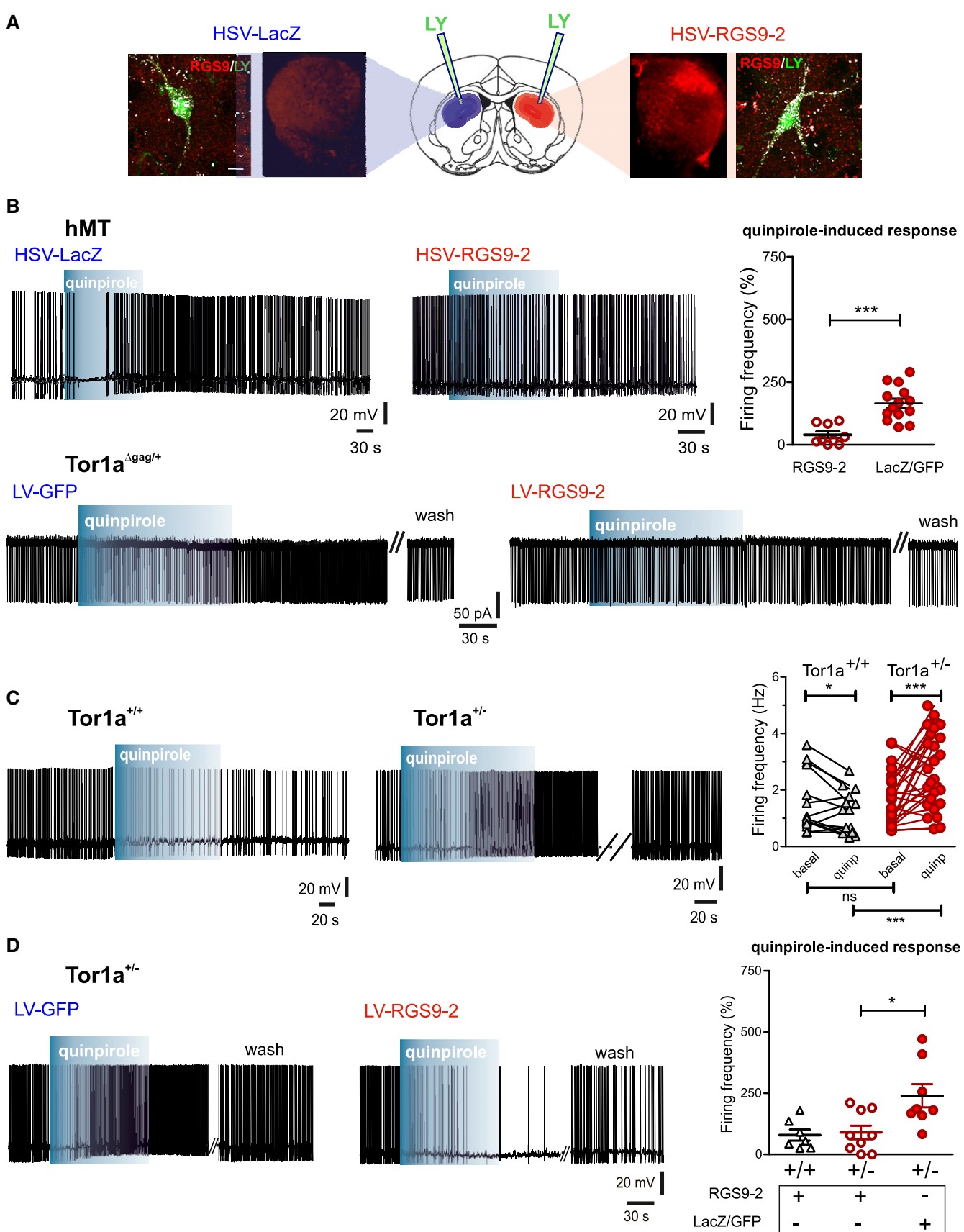

Figure 8.

# Discussion

GPCRs are central mediators of neurotransmission and are proven therapeutic targets for disease (Hauser et al, 2017). The mechanisms by which cells control GPCR trafficking and signaling are therefore topics of intensive study, and a large number of interacting pathways and partners have been uncovered. However, it is also clear that these do not yet represent the full picture of GPCR regulation. This also includes our relatively weak understanding of how the striatal D2DR is regulated, despite the fact that this receptor is central to movement control and its dysfunction is a key event in the pathogenesis of several neurological and psychiatric diseases.

*DYT1* dystonia is an incurable movement disorder, strongly associated with abnormal striatal dopaminergic responses, but its symptoms are nonresponsive to dopaminergic drugs. We show *in vivo* that: In wild-type striatum, changes in DRD2 and RGS9-2 levels are correlated during postnatal development, and RGS9-2 silencing causes DRD2 downregulation; in *DYT1* striatum: (i) DRD2 downregulation is determined by an altered receptor stability and mediated by lysosomal degradation; (ii) accordingly, changes in endosomal and autophagy–lysosomal markers support an enhanced DRD2 trafficking through the degradative pathway; (iii) reduced spinophilin and RGS9-2 levels favor β-Arr2-mediated actions; (iv) hence, RGS9-2 upregulation rescues DRD2 level and function.

Our data demonstrate in wild-type striatum a parallel increase in DRD2 and RGS9-2 protein level during postnatal development, further pointing to their close functional relationship. Though in $Tor1a^{+/-}$ mice the two proteins undergo a similar increase during early postnatal development (P7–P21), however, both exhibit a simultaneous abrupt reduction at P60. Notably, the comparable reduction of DRD2 and RGS9-2 observed in the striatum of $Tor1a^{+/-}$ and $Tor1a^{\Delta gag/+}$ mice strongly suggests that these alterations are caused by torsinA loss of function (Torres et al, 2004; Goodchild et al, 2005). Indeed, we found similarly reduced torsinA levels in P60 $Tor1a^{+/-}$ and $Tor1a^{\Delta gag/+}$ striatum, in accordance with the notion that the Δgag mutation destabilizes torsinA protein (Giles et al, 2008).

Surprisingly, our data demonstrate that different mechanisms underlie the similar reduction of DRD2 and RGS9-2 levels. Wild-type DRD2 is a stable long-lived protein, whereas RGS9-2 possesses specific degradation determinants targeting the protein for constitutive lysosomal breakdown (Hara et al, 2006). In $Tor1a^{+/-}$ mice, we found a decreased DRD2 protein half-life, whereas the sensitivity of RGS9-2 protein to lysosomal degradation was unaffected. Indeed, our data show that the reduction of RGS9-2 level in mutant mice is attributable to an enhanced localization to the DRM, in line with the increased level of its specific membrane anchor R7BP. The observation that striatal levels of a different R7 RGS family member, RGS7, and of the RGS9-2 binding partner Gβ5 are unaffected further rules out a generalized dysregulation of protein turnover in *DYT1* striatum.

Functional or structural modifications may cause a selective reduction of DRD2 protein stability in $Tor1a^{+/-}$ mice. Integral membrane proteins with limited structural/conformational defects, such as an altered post-translational modification, may escape the ER and reach the plasma membrane. This seems to be the case of DRD2 in $Tor1a^{+/-}$ striatum, since our confocal analysis shows the absence of co-localization of DRD2 and PDI signals. Commonly, these proteins are in turn identified and targeted for lysosomal proteolysis by the quality control system (Tansky et al, 2007; Apaja et al, 2010). Interestingly, modifications at the N-terminal or C-terminal regions of DRD2, altering the level of post-translational modifications, extensively affect receptor internalization rate, plasma membrane expression, and protein stability (Cho et al, 2012; Ebersole et al, 2015). Alternatively, a failure in the developmental switch from the hypersensitive "juvenile" activity to the mature state of reduced sensitivity (Kim et al, 2002; McDougall et al, 2015) may cause an increased rate of desensitization of DRD2 through the endocytic-degradative pathway after P21. The increase in Rab4 protein level observed in $Tor1a^{+/-}$ striatum is indeed suggestive of an accelerated trafficking of the receptor through the constitutive recycling pathway, which acts as a quality control system, to redirect internalized receptors toward either the plasma membrane or the degradation pathways (Apaja et al, 2010; Li et al, 2012). Endosomal recycling depends on arrestin-dependent receptor endocytosis from the plasma membrane (Kim et al, 2001; Zheng et al, 2016). Though we did not observe changes in the level of β-Arr2 in $Tor1a^{+/-}$ striatum, we found a significant decrease in spinophilin, a scaffolding protein known to interact with DRD2 (Smith et al, 1999) and to antagonize β-Arr2-dependent signaling and trafficking of GPCRs (Wang et al, 2004). Thus, spinophilin downregulation would favor β-Arr2-dependent internalization and signaling of DRD2 in $Tor1a^{+/-}$ striatum. Further, the observation that RGS9-2 co-immunoprecipitates with β-Arr2 and inhibits DRD2 internalization *in vitro* indicates antagonistic actions between RGS9-2 and β-Arr2 as well (Celver et al, 2010; Zheng et al, 2011). Our *in vivo* data strongly support this notion. Indeed, we demonstrate that RGS9-2 silencing causes DRD2 downregulation in wild-type striatum, while restoration of striatal RGS9–2 level in $Tor1a^{+/-}$ and $Tor1a^{\Delta gag/+}$ striatum rescues both protein expression and signaling of the receptor.

Thus, we hypothesize that a genetic modification resulting in torsinA loss of function might alter DRD2 maturation in the ER, causing disruption of the receptor signaling complex at the plasma membrane. The ensuing increased interaction of the receptor with β-Arr2 in turn leads to an increased DRD2 internalization rate, redirecting receptor either to a variety of alternative signaling pathways (Beaulieu & Gainetdinov, 2011) or to degradation. In support of this view, uncoupling of DRD2 from its cognate G-proteins has been reported in a *DYT1* mouse model (Napolitano et al, 2010). Our autoradiography data showing a 15% reduction of DRD2 binding density in $Tor1a^{+/-}$ striatal sections are in striking resemblance with DRD2 binding reduction reported in caudate and putamen of *DYT1* mutation carriers (Asanuma et al, 2005) and support the hypothesis of an enhanced degradation of the receptor, in accordance with previous data from different *DYT1* dystonia models (Yokoi et al, 2011; Dang et al, 2012). This hypothesis is further supported by our confocal microscopy analysis, ruling out DRD2 retention into the ER, and by immunoblotting experiments showing an increase in markers of the autophagy–lysosomal pathway, which operates constitutively at low rate to ensure quality control and turnover of long-lived proteins (Hara et al, 2006; Komatsu & Ichimura, 2010; Johansen & Lamark, 2011; Lilienbaum, 2013). In particular, we observed an increased lysosomal turnover of LC3-II (Tanida et al, 2005) in $Tor1a^{+/-}$ striatum. The observation that changes in basal LC3-II,

p62, and LAMP-2 levels did not reach statistical significance rules out an overt dysregulation of autophagy and supports a more confined impairment involving targeting of DRD2 to degradation. In line with this view, the mCherry-GFP-LC3 reporter (Castillo *et al*, 2013) was increased both in the autolysosomes and in other compartments of ChIs (Matus *et al*, 2014).

Our findings may explain why, despite a clear DRD2 involvement, dopaminergic drugs do not provide clinical benefit in *DYT1* dystonia. Furthermore, they suggest that strategies targeting β-Arr2-biased signaling or DRD2 interacting proteins, like RGS9-2, can effectively rescue striatal DRD2 function.

Alterations of DRD2-mediated dopaminergic transmission have been implicated in different neurologic and neuropsychiatric disorders. Interestingly, the clinical efficacy of antipsychotics and mood-stabilizing drugs has been ascribed to blockade of β-Arr2-biased signaling of DRD2 (Peterson & Luttrell, 2017). Furthermore, a recent study demonstrated an ameliorating effect of β-Arr2 upregulation on levodopa-induced dyskinesia in pre-clinical animal models (Urs *et al*, 2015). On the other hand, manipulation of RGS9-2 levels in animal models of levodopa-induced or tardive dyskinesia was shown to affect the manifestation of these abnormal movements (Kovoor *et al*, 2005; Gold *et al*, 2007).

With such premises, β-Arr2 and RGS9-2, as well as DRD2-biased drugs, have been proposed as drug targets (Traynor *et al*, 2009; Peterson & Luttrell, 2017; Urs *et al*, 2015; Sjögren, 2017). Accordingly, RGS9 has now been added to the IUPHAR/BPS Guide to Pharmacology database of drug targets. Such novel therapeutic approaches hold great promise for an effective treatment of *DYT1* dystonia.

# Materials and Methods

## Animal model and tissue preparation

C57BL/6, $Tor1a^{+/-}$, $Tor1a^{\Delta gag/+}$ (Goodchild *et al*, 2005), and hMT (Sciamanna *et al*, 2012b) mouse strains were bred at Fondazione Santa Lucia Animal Facility. Mice were housed under a controlled 12-h light/12-h dark cycle and with free access to food and water. DNA was isolated and amplified from 1- to 2-mm tail fragments with the Extract-N-Amp Tissue polymerase chain reaction (PCR) kit (XNAT2 kit; Sigma-Aldrich), and genotyping performed as previously described (Sciamanna *et al*, 2012b; Ponterio *et al*, 2018). Mice were sacrificed by cervical dislocation and brains quickly removed from the skull. Preparation of corticostriatal slices was carried out as previously described (Sciamanna *et al*, 2011, 2012b; Ponterio *et al*, 2018). Briefly, coronal slices (210 μm) were cut with a vibratome in oxygenated Krebs' solution (in mM): 126 NaCl, 2.5 KCl, 1.3 $MgCl_2$, 1.2 $NaH_2PO_4$, 2.4 $CaCl_2$, 10 glucose, 18 $NaHCO_3$). After 30-min recovery, a single slice was transferred to the recording chamber, on the stage of an upright microscope (BX51WI, Olympus), submerged in continuously flowing oxygenated (95% $O_2$/5% $CO_2$) Krebs' solution (2.5–3 ml/min). ChIs were visualized by means of IR-DIC videomicroscopy (C6790 CCD, Hamamatsu).

## Experimental design

Age- and sex-matched wild-type and mutant mice were randomly allocated to experimental groups and processed by a blinded investigator. Sample size for any measurement was based on the ARRIVE recommendations on refinement and reduction of animal use in research, as well as on our previous studies. For data analysis of WB, confocal and binding images, and electrophysiology experiments, the investigator was blinded to genotype/treatment. Each observation was obtained from an independent biological sample. The number of biological replicates is represented with *N* for number of animals and *n* for number of cells. For electrophysiology, each cell was recorded from a different brain slice.

## Immunoblotting

WB of striatal lysates was performed as previously described (Napolitano *et al*, 2010). Mouse striata were homogenized in cold buffer: 50 mM Tris–HCl pH 7.4, 150 mM NaCl, 1% Triton X-100, 0.25% Na deoxycholate, 5 mM $MgCl_2$, 0.1% SDS, 1 mM EDTA, and 1% protease inhibitor cocktail (Sigma-Aldrich). Samples were sonicated and kept on ice for 1 h. Then, crude lysates were centrifuged (15,600 *g*, 15 min, 4°C), the supernatant collected, and protein quantified with Bradford assay (Bio-Rad). Protein extracts (15–30 μg) were loaded with NuPAGE LDS sample buffer (Invitrogen, Life Technologies Italia), containing DTT. Samples were denatured (95°C, 5 min) and loaded onto 8–15% SDS–PAGE gels. Gels were blotted onto 0.45-μm polyvinylidene fluoride (PVDF) membranes. The following primary antibodies were used (Appendix Table S1): guinea pig anti-p62 (1:300, GP62-C; Progen), mouse anti-LC3 (1:250, 0231-100; NanoTools), goat anti-RGS9 (1:1,000, sc-8142; Santa Cruz), rabbit anti-DRD2 (1:1,000, AB5084P; Millipore), rabbit anti-torsinA (1:800, AB34540; Abcam), goat anti-RGS7 (1:200, sc-8139; Santa Cruz), rabbit anti-Gβ5 (1:1,000, 071208; Millipore), goat anti-R7BP (1:500, sc-50170; Santa Cruz), rabbit anti-Rab4 (1:1,000, ab13252; Abcam), rabbit anti-spinophilin/neurabin-II (1:3,000, 5162; Sigma-Aldrich), rat anti-LAMP-2 (1:200, GTX13524; Genetex), mouse anti-β-arrestin-2 (1:500, sc-13140; Santa Cruz), rabbit anti-GFP (1:200, sc-8334; Santa Cruz), 1 overnight at 4°C; mouse anti-β-actin (1:20,000, A5441; Sigma-Aldrich), mouse anti-β-tubulin (1:20,000, T4026; Sigma-Aldrich), mouse anti-PSD-95 (1:20,000, MAB1598; Millipore), 30–60 min at RT. Anti-goat, anti-guinea pig, anti-mouse, anti-rat, and anti-rabbit horseradish peroxidase (HRP)-conjugated secondary antibodies were used (GE Healthcare). Immunodetection was performed by ECL reagent (GE Healthcare), and membrane was exposed to film (Amersham). Quantification was achieved by ImageJ software (NIH). Detergent-resistant membrane fractions were prepared according to Celver and coll. (Celver *et al*, 2012). Treatment of slices with bafilomycin A₁ (BafA₁, 5 nM, 1 h) was performed as previously described (Dehay *et al*, 2012). For the protein degradation assay, slices were incubated for 5 h in oxygenated ACSF at 32°C with or without leupeptin (100 μM; Anderson *et al*, 2007a). After treatments, slices were immediately lysed for immunoblotting, as described. Sample exclusion criteria were loading control signal saturation or high background.

## Receptor autoradiography

The radioligand binding protocol has been modified from Fasano *et al* (2009). Briefly, fresh-frozen dissected mouse brains were sliced coronally at 16 μm on a cryostat at the level of the caudate–putamen (plate 15–36 of Franklin & Paxinos, 2008). Unfixed slides were preincubated

in assay buffer for 60 min [50 mM Tris–Cl (pH 7.4), 120 mM NaCl, 5 mM KCl, 1 mM $MgCl_2$, and 40 nM ketanserin]. Sections were then incubated for 60 min at room temperature in the same buffer with the addition of 5 nM $^3$H-spiperone (PerkinElmer, Boston, MA) and 40 nM ketanserin. Cold competition of tritiated spiperone with 10 μM cold spiperone was used to assess nonspecific signal. After ligand binding, slides were rinsed twice for 10 min in the same buffer, allowed to dry, and exposed to Hyperfilm MP film (Merck) for 4 weeks.

### Image analysis and quantification

Quantification analyses were performed in blind, and sample identity was not revealed until correlations were completed. Densitometric analyses and ROD measurements were performed as previously described (Pratelli et al, 2017). Briefly, 10 sections per animal along the whole rostro-caudal extent of the striatum were used. Optical density (OD) was evaluated in the caudate–putamen, and background OD value was determined in structures of the same section devoid of specific signal and subtracted for correction to obtain the relative OD (ROD) value. Results were expressed as percentage increase/decrease in $^3$H-spiperone density. Data were analyzed by Student's t-test.

### Confocal imaging

Slices processing and confocal image acquisition were performed as previously described (Vanni et al, 2015; Ponterio et al, 2018). Mice were deeply anesthetized and perfused with cold 4% paraformaldehyde in 0.12 M phosphate buffer (pH 7.4). The striatum was dissected, post-fixed for at least 3 h at 4°C and equilibrated with 30% sucrose overnight. Coronal striatal sections (30 μm thick) were cut with a freezing microtome. Slices were dehydrated with serial alcohol dilutions (50–70–50%) and then incubated 1 h at RT in 10% donkey serum solution in PBS 0.25%-Triton X-100 (PBS-Tx). The following primary antibodies were utilized (3 days at 4°C; Appendix Table S2): goat anti-ChAT (1:500, NBPI30052; Novus Biologicals); rabbit anti-DRD2 (1:500, AB5084P; Millipore); goat anti-RGS9 (1:1,000, sc-8142; Santa Cruz), mouse anti-PDI (1:500, SPA891; StressGene); guinea pig anti-P62 (1:300, GP62-C; Progen). The following secondary antibodies were used (1:200, RT, 2 h): Alexa 488 and Alexa 647 (Invitrogen), and cyanine 3 (cy3)-conjugated secondary antibodies (Jackson ImmunoResearch). After washout, slices were mounted on plus polarized glass slides with Vectashield mounting medium (Super Frost Plus; Thermo Scientific) and coverslipped. Images were acquired with a LSM700 Zeiss confocal laser scanning microscope (Zeiss, Germany), with a 5×, a 20× objective, or a 63× oil immersion lens (1.4 numerical aperture) with an additional digital zoom factor (1×–1.5×–2×). Single-section images (1,024 × 1,024) or z-stack projections in the z-dimension (z-spacing, 1 μm) were collected. Z-stack images were acquired to analyze the whole neuronal soma, which spans multiple confocal planes. The confocal pinhole was kept at 1, the gain and the offset were adjusted to prevent saturation of the brightest signal and sequential scanning for each channel was performed. The confocal settings, including laser power, photomultiplier gain, and offset, were kept constant for each marker. For quantitative analysis, images were collected from at least 3–4 slices processed simultaneously from each striatum (n ≥ 3 mice/genotype) and exported for analysis with ImageJ software (NIH). Software background

subtraction was utilized to reduce noise. To quantify the density of a specific marker on a defined area, we calculated the "overlapping signal" by utilizing a region of interest (ROI), as previously described (Vanni et al, 2015). When neurons were loaded with lucifer yellow (0.2%) through the patch pipette during the electrophysiological recordings, slices were fixed overnight with 4% paraformaldehyde in 0.12 M phosphate buffer (pH 7.4), washed in PBS, mounted with Vectashield mounting medium on plus polarized glass slides (Super Frost Plus Thermo Scientific), and coverslipped (Ponterio et al, 2018).

### Viral constructs and preparations

For RGS9-2 silencing by lentiviral (LV) particles, two candidate shRNA sequences were designed to specifically silence mouse RGS9-2 gene (NCBI GenBank AF125046), according to the clone sequences provided by Sigma-Aldrich (clones shRNA_RGS9-2: 37134 e 37135). Each sequence was inserted into the miR30a backbone sequence (formed by the stem loop and by 100-bp downstream and upstream the premiR30a) to increase the expression efficiency of the shRNA (Denti et al, 2004).
shRNA37134:    TTATCTTTTCCACCCAAGCTTTGTAAAAATAAATC AAAGAGAAAGCAAGGTATTGGTTTCAGCCAACAAGATAATTACTC CCTTGAAGTTGGAGGCAGTAGGCACCCAAGTGCATTAGGATAATA CCCATTTGTGGCTTCACAGTATTATCCTAATGCACTTGGGGTCGCT CACTGTCAACGTTGATATGCCTTCTTCAGCATTCTGTCTTACTGAC CTGAGAAGTGCTCTGCGGGAGTTTCTGAAATGTACAGGCAACATT CTGTAAAC
shRNA37135: GTTTACAGAATGTTGCCTGTACATTTCAGAAACTCC CGCAGAGCACTTCTCAGGTCAGTAAGACAGAATGCTGAAGAAGGC ATATCAACGTTGACAGTGAGCGACCCGATTTCAGACGCCATATTTC TGTGAAGCCACAAATGGGAAATATGGCGTCTGAAATCGGTGCCTA CTGCCTCCAACTTCAAGGAGTAATTATCTTGTTGGCTGAAACCAAT ACCTTGCTTTCTCTTTGATTTATTTTTACAAAGCTTGGGTGGAAAA GATAA

The sequences were cloned into the p207.pRRLsinPPTs.hCMV. GFP.WPREp207 plasmid (kindly provided by L. Naldini, Milan, Italy) under the U1 promoter for shRNA expression (p207 was modified as p207-U1: p207_U1 shRNA-34 and p207_U1 shRNA-35) upstream to the CMV promoter. Both plasmids carrying each sequence were used for a single LV preparation. After cloning, the VSV-G-pseudotyped lentiviral vectors were generated by calcium phosphate transfection of HEK293T cells with a mixture of the four plasmids required to produce third-generation lentiviruses (kindly provided by L. Naldini, San Raffaele Scientific Institute, Milan, Italy). The LV particles were prepared and purified according to previously published protocols (Mandolesi et al, 2009; Sciamanna et al, 2012b).

For overexpression of RGS9-2 by LV particles, the RGS9-2 cDNA sequence (NCBI GenBank AF125046) was cloned in the p207.pRRLsinPPTs.hCMV.WPREp207 under the CMV promoter. After cloning, the VSV-G-pseudotyped lentiviral vectors were prepared as described above.

The herpes simplex viral (HSV) vector used for overexpressing mouse RGS9-2 (NCBI GenBank AF125046) has been previously described and validated (Rahman et al, 2003; Gold et al, 2007). The description of the control HSV-LacZ (β-galactosidase) vector is provided elsewhere (Carlezon et al, 1997). After cloning, HSV constructs were grown and purified according to previously published protocols (Coopersmith & Neve, 1999; Carlezon et al, 2000).

To monitor *in vivo* autophagy flux in ChIs, we utilized an adeno-associated virus (AAV2/9) carrying a monomeric tandem mCherry-GFP-LC3 construct (AAV2/9-mCherry-GFP-LC3; kindly provided by C. Hetz) as a reporter (Castillo *et al*, 2013). The GFP fluorescent signal of the reporter is sensitive to acidic conditions; thus, co-localization of green and red fluorescence (yellow puncta) indicates that the tandem protein is not localized in compartments fused with a lysosome, while detection of red puncta indicates that the protein is located in the autolysosome (Matus *et al*, 2014).

**Stereotactic injection of viral particles**

Viral injection into the dorsal striatum was performed as previously described (Mandolesi *et al*, 2009; Fasano *et al*, 2010; Sciamanna *et al*, 2012b; Bourdenx *et al*, 2015; Urs *et al*, 2015). Briefly, male $Tor1a^{+/-}$ and $Tor1a^{\Delta gag/+}$ mice were anesthetized with tiletamine/zolazepam (80 mg/kg) and xylazine (10 mg/kg). Viral preparation suspensions were injected into the dorsal striatum, using a glass capillary (Sutter Instruments) connected to a picospritzer (Parker Inst, USA) as previously described (e.g., see Mandolesi *et al*, 2009; Fasano *et al*, 2010; Sciamanna *et al*, 2012b; Bourdenx *et al*, 2015; Urs *et al*, 2015). The following coordinates from Bregma were used for dorsal striatum: anteroposterior +0.4 mm; lateral ±2.5 mm; and ventral at −1.7 and −2 mm from the dura. Mice received 2 μl of viral preparation administered at 0.1 ml/min rate. At the end of the injection, the capillary was left in place for 5 min before being slowly removed. The skin was sutured, and mice were allowed to recover. In line with preliminary immunofluorescence experiments assessing the time-course and extent of viral infection, mice were sacrificed either 3–5 days after injection of HSV particles, or 3–4 weeks following LV or AAV particles.

**Electrophysiology**

Electrophysiological patch-clamp recordings were performed as previously described from individual ChIs in striatal coronal slices, prepared as previously described from P60 to P90 old mice (Sciamanna *et al*, 2011, 2012b; Ponterio *et al*, 2018). ChIs were visualized using standard IR-DIC microscopy and identified based on their large somatic size and distinctive electrophysiological properties. Electrophysiological signals were detected using Multiclamp 700B and AxoPatch 200 amplifiers (Molecular Devices), using borosilicate glass pipettes pulled on a P-97 Puller (Sutter Instruments). For cell-attached recordings, the electrodes were filled with a solution containing the following (in mM): 140.5 $KMeSO_4$, 0.2 EGTA, 7.5 NaCl, 10 HEPES, 2 NaATP, and 0.2 NaGTP, adjusted to a pH of 7.3 with KOH. For perforated-patch recordings, gramicidin was added at a final concentration of 20 μg/ml and the perforation process was considered complete when both the amplitude of the action potentials and electrode resistance were steady. Action potential firing frequency was analyzed with Clampfit 10 (pClamp 10, Molecular Devices); neurons with frequencies outside the 0.5–4.5 range were excluded from further analysis.

**Statistics**

All data were obtained from at least two independent experiments. Sample size for any measurement was based on the ARRIVE

**The paper explained**

**Problem**

DYT1 dystonia is a progressive and highly disabling disease, with symptom onset frequently between childhood and adolescence. DYT1 causative mutation has been identified in the *Tor1a* gene. However, the molecular mechanisms underlying symptom manifestation are far from being clarified and medical treatments are unavailable. Dysfunction of dopamine D2 receptors in the striatum, a brain region involved primarily in motor control and motor learning, has been reported in DYT1 patients and rodent models. Therefore, restoring striatal D2 receptor function represents a potential therapeutic strategy.

**Results**

We investigated the molecular mechanisms of striatal dopamine D2 receptor dysfunction in multiple mouse models of DYT1 dystonia. We found that the striatal levels of D2 receptor, and of its regulatory proteins spinophilin and RGS9-2, are reduced. We show that D2 receptor downregulation is mediated by an abnormal, selective trafficking to lysosomal degradation. Furthermore, we present evidence that viral-induced expression of RGS9-2 is able to restore both the striatal level and the function of dopamine D2 receptor.

**Impact**

These results provide an explanation for the lack of effectiveness of dopaminergic drugs in DYT1 dystonia, despite a clear involvement of dopamine receptors in disease pathophysiology. More importantly, our work identifies therapeutic targets that are effective in rescuing the dopaminergic dysfunction in DYT1 dystonia.

recommendations on refinement and reduction of animal use in research, as well as on our previous studies. Age- and sex-matched wild-type and mutant mice were randomly allocated to experimental groups and processed by a blinded investigator. For data analysis of WB, confocal and binding images, and electrophysiology experiments, the investigator was blinded to genotype/treatment. Each observation was obtained from an independent biological sample. The number of biological replicates is represented with $N$ for number of animals and $n$ for number of cells. For electrophysiology, each cell was recorded from a different brain slice. Data analysis was performed with Clampfit 10 (pClamp 10), ImageJ (NIH), and Prism 5.3 (GraphPad). Data are reported as mean ± SEM. Statistical significance was evaluated as indicated in the text, with Pearson's $r$ correlation test, one-way ANOVA with *post hoc* tests between groups corrected for multiple comparisons, and two-tailed two-sample $t$-test (parametric or nonparametric, unpaired, or paired) as appropriate according to each test assumptions. For example, normality tests were used to assess Gaussian distribution. $F$-test was used to compare variances between groups; when variance was different, Welch's correction was used. Statistical tests were two-tailed, the confidence interval was 95 %, and the alpha-level used to determine significance was set at $P < 0.05$.

**Study approval**

Animal breeding and handling were performed in compliance with the ethical and safety rules and guidelines for the use of animals in biomedical research provided by the European Union's directives and Italian laws (2010/63EU, D.lgs. 26/2014; 86/609/CEE, D.Lgs 116/1992). The experimental procedures were approved by

Fondazione Santa Lucia Animal Care and Use Committee and the Italian Ministry of Health (authorization # 223/2017-PR).

Expanded View for this article is available online.

## Acknowledgements

We are grateful to Dr. Georgia Mandolesi for the lentiviral preparations and to Mr. Massimo Tolu and Mr. Vladimiro Batocchi for their excellent technical assistance. We wish to thank Prof. Claudio Hetz for kindly providing AAV2/9-mCherry-GFP-LC3, Prof. L. Naldini (San Raffaele Scientific Institute, Milan, Italy) for kindly providing the lentiviral plasmids, Dr. Nutan Sharma and Prof. David Standaert for kindly providing the hMT mice. We are also grateful to Prof. Xandra O. Breakefield for critical reading of the manuscript, Prof. Emiliana Borrelli and Dr. Marcello Serra for critical discussion and providing samples for DRD2 antibody validation, and Dr. Tilmann Achsel for valuable technical suggestions. A.P. and R.E.G. are grateful for the support and effort of the Dystonia Medical Research Foundation and the Foundation for Dystonia Research. This work was supported by grants to A.P. from Dystonia Medical Research Foundation (DMRF, 2011 Stanley Fahn Award) and the Italian Ministry of Health, Ricerca Finalizzata (RF-2010-2311657), and to M.D.A. from the Italian Ministry of Health (Progetto Giovani Ricercatori GR-2011-02351457) and from the Alzheimer's Association, United States (AARG-18-566270).

## Author contributions

PB and AP conceived and designed the study. GP, VV, AT, GS, GM, MM, SM, and ED performed the experiments and analyzed the data. PB supervised data analysis, evaluated the results, and wrote the manuscript. GP, VV, and AT contributed to critical evaluation of the results. BD, VZ, REG, MDA, MP, and EB provided experimental resources and contributed to critical evaluation of the results. AP, BD, REG, NBM, MDA, MP, and EB edited the manuscript. All authors approved the manuscript.

## Conflict of interest

The authors declare that they have no conflict of interest.

## For more information

(i)   OMIM #128100—DYSTONIA 1, TORSION, AUTOSOMAL DOMINANT; DYT1
      http://omim.org/entry/128100

(ii)  MDS Gene overview of studies for DYT-TOR1A
      http://www.mdsgene.org/d/4/g/14?fc=0&_mu=1

(iii) gene #1861—TOR1A, torsin family 1 member A [*Homo sapiens*]
      https://www.ncbi.nlm.nih.gov/gene/1861

(iv)  gene #30931—Tor1a, torsin family 1, member A (torsin A) [*Mus musculus*]
      https://www.ncbi.nlm.nih.gov/gene/30931

(v)   gene #19739—Rgs9, regulator of G protein signaling 9 [*Mus musculus*]
      https://www.ncbi.nlm.nih.gov/gene/19739

(vi)  gene #13489—Drd2, dopamine receptor D2 [*Mus musculus*]
      https://www.ncbi.nlm.nih.gov/gene/13489

(vii) Dystonia Europe: https://dystonia-europe.org/

(viii) Dystonia Coalition: https://www.rarediseasesnetwork.org/cms/dystonia

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
