## [Review Process File · EMBO Molecular Medicine]

RGS9-2 rescues dopamine D2 receptor levels and signaling in DYT1 dystonia mouse models

Paola Bonsi, Giulia Ponterio, Valentina Vanni, Annalisa Tassone, Giuseppe Sciamanna, Sara Migliarini, Giuseppina Martella, Maria Meringolo, Benjamin Dehay, Evelyne Doudnikoff, Venetia Zachariou, Rose E. Goodchild, Nicola B. Mercuri, Marcello D'Amelio, Massimo Pasqualetti, Erwan Bezard, and Antonio Pisani

Review timeline:

Submission date:	2 May 2018
Editorial Decision:	8 June 2018
Revision received:	9 October 2018
Editorial Decision:	6 November 2018
Revision received:	21 November 2018
Accepted:	23 November 2018

Editor: Céline Carret

Transaction Report:

1st Editorial Decision

8 June 2018

Thank you for the submission of your manuscript to EMBO Molecular Medicine. We have now heard back from the three referees whom we asked to evaluate your manuscript. As you will see from the reports below, the referees find the topic of your study of potential interest. However, they raise substantial concerns on your work, which should be convincingly addressed in a major revision of the present manuscript.

In particular, ref. 1 and 2 have similar serious concerns about the antibody used and request convincing validation of D2 reactivity, ideally on D2 KO animals. Further experiments and clarifications are requested as well to improve conclusiveness and make the paper more accessible to non-experts.

I realize that the antibody issue is a serious but tricky one. Unfortunately, we feel that addressing this satisfactorily is a condition for the paper to move forward. I am uncertain whether you will be able (or willing) to return a revised manuscript within the 3 months deadline under this condition, and I would also understand your decision if you chose to rather seek rapid publication elsewhere at this stage.

This said, we would welcome the submission of a revised version within three months for further consideration and would like to encourage you to address all the criticisms raised as suggested to improve conclusiveness and clarity. Please note that EMBO Molecular Medicine strongly supports a single round of revision and that, as acceptance or rejection of the manuscript will depend on another round of review, your responses should be as complete as possible.

I look forward to receiving your revised manuscript.

Should you find that the requested revisions are not feasible within the constraints outlined here and choose, therefore, to submit your paper elsewhere, we would welcome a message to this effect.

***** Reviewer's comments *****

Referee #1 (Comments on Novelty/Model System for Author):

The medical impact will be very high if the authors' conclusions are valid. However, lack of validation of the antibody reagents used in the study make any of the conclusions in the study highly questionable.

In addition the clarity of the writing needs to be improved to make it interesting to the non-specialist.

Referee #1 (Remarks for Author):

Review of "RGS9-2 rescues dopamine D2 receptor levels and signalling in DYT1 dystonia mouse models".

There are serious concerns about the validity of the conclusions drawn in this manuscript primarily because of lack of validation of the antibodies used by the investigators to detect and quantify the endogenously expressed proteins. In particular this reviewer is concerned that the D2 dopamine receptor (DRD2) signal that the authors detect and measure in their investigation is not from DRD2. If their antibody is not really detecting DRD2 then the major conclusions draw by the authors are not valid.

In Fig. 1 a1 the authors show an immunoblot of DRD2 using a commercially available antibody from Millipore. These immunoblots were utilized to quantify the levels of D2RD2 in the brains of mice. This reviewer has strong concerns about the use of validity of the antibody and the single DRD2 band shown in the immunoblot. The concerns are based on almost 20 years of experience that this reviewer has had studying the biochemistry of DRD2.

The major concern is that the authors do not provide any data proving that the antibody is indeed recognizing DRD2 in the brain.

The antibody used is AB5084P from Millipore, a rabbit polyclonal antibody generated against a peptide corresponding to residues in the 3rd cytoplasmic loop of DRD2. However, polyclonal antibodies are notoriously non-specific and could non-selectively and non-specifically target numerous brain proteins. The extremely poor reliability of the vast number of anti-GPCR antibodies for recognizing their target GPCR has been documented in the following manuscript: Naunyn-Schmiedeberg's Archives of Pharmacology, April 2009, Volume 379, Issue 4, pp 385-388. "How reliable are G-protein-coupled receptor antibodies?" In fact, it has been shown with anti-GPCR antibodies that "previously applied validation of such antibodies by the disappearance of staining in the presence of blocking peptide, i.e. the antigen against which the antibody was raised, alone is insufficient to demonstrate specificity".

This reviewer has had much experience evaluating and validating many of these supposedly anti-DRD2 antibodies that are cited quite commonly in the literature including AB5084P, which was originally provided by Chemicon. Many of these antibodies fail to detect bands that are specific to D2DR even when the protein is transiently expressed at high levels in cell lines, i.e. there are no bands detected that are unique to cells overexpressing D2RD2 compared to untransfected cells. The

problem becomes even more severe when trying to detect DRD2 endogenously expressed in brain. DRD2, in particular among GPCRs, is expressed at very low levels in brain and highly insoluble in detergents. Furthermore, even under highly denaturing conditions with urea and SDS in the sample buffer much of the DRD2 signal fails to penetrate into the PAGE gel.

Furthermore, this reviewer has attempted to validate these commercial antibodies, including AB5084P, using brain tissue from D2RD2 knockout mice. While multiple protein bands are detected, some corresponding to the predicted molecular weight of DRD2, bands specific to wild-type mice that are not present in the knockout mouse tissue are never detected.

The authors of the above-cited manuscript, "How reliable are G-protein-coupled receptor antibodies?" indicate that data from GPCR antibodies should only be considered valid if the following can be demonstrated: a) disappearance of staining in knock-out animals of the target receptor, b) reduction of staining upon knock-down approaches such as siRNA treatment. This reviewer concurs and further believes that such validation should be compulsory if investigators are trying to detect and quantify endogenously expressed DRD2. Many anti-GPCR antibodies can recognize the target GPCR when it is overexpressed in cell line but only exhibit non-specific staining *in vivo*. If signal from these antibodies is sufficiently amplified most polyclonal antibodies will identify protein bands close to any molecular weight. In addition in immunohistochemistry experiments the antibody will show a pattern of staining but much or all of that staining could be non-specific.

b) In their immunoblots the investigators show DRD2 running as a clean single band. This reviewer has never observed DRD2 running in this manner. The reviewer has worked with an anti-DRD2 antibody was validated against DRD2 knockout tissue and DRD2 from brain runs as multiple bands starting at around 60kD as the lowest and then extending to multiple higher bands. Moreover there is strong smeared signal between the more prominent bands that extends into the stacking gel.

In summary, while the anti-DRD2 antibody is referenced in multiple publications on DRD2 there is no proof that it recognizes DRD2, especially endogenously expressed DRD2.

The lack of validation extends to some of the other antibodies used by the investigators. The investigators need to clearly describe how the antibodies were validated. Simply showing that the antibodies detect a band running at the right molecular weight is not sufficient. The authors need to provide documentation (their own or from the supplier) showing that the antibody detects the appropriate signal in tissue known to express the target protein and that there is no signal in tissue that does not express the protein. Alternatively, multiple antibodies targeting different epitopes of the target protein should be evaluated to ensure that the similar results are obtained with each of the different antibodies.

Referee #2 (Remarks for Author):

The article by Bonsi et al. investigates on the possible mechanisms responsible for the reduction of D2 receptor signalling in Dystonia using *Dyt1* mice. Authors report that the reduction depends on increased lysosomal degradation of the D2 receptor due to reduction of proteins interacting with it. In particular, they report on the critical interaction between D2/RGS9-2 proteins, which regulates the expression and localization of these proteins. They also show that viral expression of RGS9-2 can rescue D2 functions in ChIs, as shown using an electrophysiological approach. Overall the findings are interesting and have translational value. Nevertheless, there are points that need to be addressed in support of their data and conclusions. In particular, D2 reduction is not convincing; additional approaches should be used to support western blot analyses. Furthermore, the antibody against D2 needs to be verified using knockout mice.

Specific points:

1) The *Dyt1* Tor1A mutation and δ gag mutation are not introduced nor it is explained how the mice were made and why only the heterozygotes are used. Readers not familiar with the disorder need this information.

2) In Fig 1 all the experiments shown are made with extracts from P60 mice while in the text is mentioned P60-P90. What was done at P90?

3) The D2 and RGS9-2 down-regulation is not convincing; less exposed western blot might be more convincing. What is the size of D2 on these western blots?

Most commercial antibodies against D2 are not specific, it would be good to run an extract from D2KO mice as control of antibody specificity. Also additional experiments should be made in support of this important point. For instance, D2 ligand binding experiments on striatal membrane would be very appropriate.

4) Similarly, if RGS9-2 is responsible for the D2 trafficking in the cells and since it is very hard to evaluate this in vivo, in vitro experiments should be performed using transfected D2 and RGS9-2 in mammalian cells. In this system it would be possible to change the ratio of the two proteins and analyze more in depth the lysosomal degradation.

5) Is D2 expression, in the brain of mice heterozygous for any of the proteins mentioned in the autophagy-lysosomal pathway, reduced? This would support results contained in fig. 3-6, where at present differences between WT and KO are not striking.

Referee #3 (Remarks for Author):

This is an interesting study that provides insight into mechanisms underlying an animal model of severe DYT1 dystonia, the TOR1A mouse. They discovered reduced striatal expression of dopamine receptor 2 (DRD2) in these mice and propose that reduced expression of proteins that counteract DRD2 internalization and degradation mechanisms (RGS9-2 and spinophilin). As DYT-1 dystonia patients respond poorly to dopaminergic drugs, these findings have significant implications for pathophysiology and treatment. Overall the data are convincing and the paper well written. There are several issues that the authors should address prior to publication.

Major comments:

1. Please provide some background on the TOR1A and TOR1a (deltaGAG) mutant mouse models, the functions torsinA, and its relationship to DRD2 signaling and dystonia.

2. Figure 1 The reductions in RGS9-2 levels in the TOR1A mouse are not impressive. It would be important to know if its obligate binding partners Gb5 or R7BP are concomitantly reduced in these experiments.

3. Figure 2. What is meant exactly by a "trend" in the reduction of DRD2 in DRMs of TOR1A striatum? Since the graph shows a wide range of DRD2 values (actually 2 distinct groups) in the +/- mutant mice, a few more mice could determine whether this is truly significant or not. Since RGS9-2 is unstable without Gb5 or R7BP, is the localization of these proteins in DRMs similarly altered?

4. Figure 3. What is meant by "sham" transfection and why wasn't a control (non-targeting) shRNA used? Figure 3e: The "trend" referred to is not apparent; either the data are significant or not-please clarify.

Minor comments:

Page 3, last paragraph "These findings can [explain] the paradox..."

1st Revision - authors' response

9 October 2018

1) Referee #1 (Comments on Novelty/Model System for Author):

The medical impact will be very high if the authors' conclusions are valid. However, lack of validation of the antibody reagents used in the study make any of the conclusions in the study highly questionable.

In addition the clarity of the writing needs to be improved to make it interesting to the non-specialist.

RE:

We wish to thank the reviewer; indeed she/he raised a central issue, the lack of medical treatment for dystonia. Indeed, Deep Brain Stimulation (DBS) or botulinum toxin injections represent the few available options for dystonia, a disabling disorder that highly compromises patients' life quality. Therefore, this represents an unmet clinical need, requiring preclinical work to be implemented. We believe in the translational impact of our work and in the novelty of the target proposed. Indeed, we assume that a post-receptor target might be a means to bypass dysfunctional transmitter receptors, thereby representing an absolute novelty for movement disorders such as dystonia.

Additionally, we took into careful consideration the Reviewer's concerns, and directly addressed them by performing a significant amount of new experiments, in attempt to convince the Reviewers and the readers of the correctness of the conclusions drawn. Yet, the text was amended to make it easier to follow for non-specialists.

2) Referee #1 (Remarks for Author):

Review of "RGS9-2 rescues dopamine D2 receptor levels and signalling in DYT1 dystonia mouse models".

There are serious concerns about the validity of the conclusions drawn in this manuscript primarily because of lack of validation of the antibodies used by the investigators to detect and quantify the endogenously expressed proteins. In particular this reviewer is concerned that the D2 dopamine receptor (DRD2) signal that the authors detect and measure in their investigation is not from DRD2. If their antibody is not really detecting DRD2 then the major conclusions draw by the authors are not valid.

RE:

In principle, we agree with the reviewer that caution is required when utilizing antibodies against GPCRs, as discussed in the cited issue of Naunyn-Schmiedeberg's Archives of Pharmacology. However, we should consider that a reduction of DRD2 receptor in DYT1 dystonia has been consistently reported by different groups with diverse methods, in humans as well as mouse models. In human DYT1 mutation carriers, DRD2 binding is reduced by 15% (Asanuma et al., 2005). DRD2 binding has been found similarly reduced in two DYT1 mouse models, a striatal-specific torsinA knockout and a knockin for the torsinA mutation (Yokoi et al., 2011; Dang et al., 2012). Accordingly, DRD2 protein expression levels have been previously found reduced both by our group and by others in different mouse models: transgenic mice overexpressing human mutant torsinA (Napolitano et al., 2010) and mice knockin for the torsinA mutation (Dang et al., 2012). Anyway, we are aware that the antibody issue need careful consideration, and tested several commercially available DRD2 antibodies (Millipore, Abcam, Santa Cruz) before choosing the Millipore AB5084P antibody. The Chemicon-Millipore AB5084P antibody was generated and thoroughly validated as described in Boundy et al., 1993. It is the most used and referenced, and has been used also for electron microscopy (Galvan et al., 2014; Wang and Pickel, 2002). More importantly, recently this antibody has been successfully validated by using both knock-out mice and immunoprecipitation with subsequent spectrometrical identification of the immunoprecipitate (Stojanovic T, et al., Validation of dopamine receptor DRD1 and DRD2 antibodies using receptor deficient mice. Amino Acids. 2017 Jun;49(6):1101-1109). This reference is now included into Appendix Table 1.

In further support of AB5084P antibody specificity, in Figure 6 of our manuscript we showed a striatal-specific DRD2 immunolabeling and absence of labeling in the cortex, a pattern that is superimposable to the specific signal shown by Stojanovic and coll. (2017) as well as Wang and Pickel (2002).

To further answer to the reviewers' concern about the DRD2 antibody specificity, we validated AB5084P antibody in our experimental setting, by utilizing D2DR knockout mouse samples provided by Prof. E. Borrelli. The figure below shows an immunoblotting with Millipore AB5084P antibody against DRD2 (Lot # 2279497) on a lysate of DRD2 knockout mouse, kindly provided by Prof. Borrelli (lane 1) and a lysate from a control mouse of our animal facility (lane 2). β -actin was

used as loading normalizer. The immunoblotting shows lack of DRD2 signal in knockout sample, with similar loading indicated by actin signal.

Additionally, we tested the Millipore AB5084P antibody by immunohistochemistry. The figure below shows DRD2 immunolabeling of the striatum from a control ED-43 mouse (A), and absence of signal in two striatal sections from a DRD2 knockout mouse brain (B, C) kindly provided by Prof. Borrelli.

3) In Fig. 1 a1 the authors show an immunoblot of DRD2 using a commercially available antibody from Millipore. These immunoblots were utilized to quantify the levels of DRD2 in the brains of mice. This reviewer has strong concerns about the use of validity of the antibody and the single DRD2 band shown in the immunoblot. The concerns are based on almost 20 years of experience that this reviewer has had studying the biochemistry of DRD2.

RE:

Please see point 2 and point 9.

4) The major concern is that the authors do not provide any data proving that the antibody is indeed recognizing DRD2 in the brain.

RE:

Please see point 2. Furthermore, during the revision of the manuscript, in collaboration with Dr. Pasqualetti (Univ. of Pisa, Italy) we performed radioligand binding experiments (included in revised Figure 1), which showed a reduction of DRD2 binding in striatal slices from mutant mice, in accordance with western blotting data.

5) The antibody used is AB5084P from Millipore, a rabbit polyclonal antibody generated against a peptide corresponding to residues in the 3rd cytoplasmic loop of DRD2. However, polyclonal antibodies are notoriously non-specific and could non-selectively and non-specifically target numerous brain proteins. The extremely poor reliability of the vast number of anti-GPCR antibodies for recognizing their target GPCR has been documented in the following manuscript: Naunyn-Schmiedeberg's Archives of Pharmacology, April 2009, Volume 379, Issue 4, pp 385-388. "How reliable are G-protein-coupled receptor antibodies?" In fact, it has been shown with anti-GPCR antibodies that "previously applied validation of such antibodies by the disappearance of staining in

the presence of blocking peptide, i.e. the antigen against which the antibody was raised, alone is insufficient to demonstrate specificity".

RE:

As discussed above (point 2), the AB084P antibody has been recently validated on knockout mice, with immunoprecipitation and spectrometrical identification of the immunoprecipitate (Stojanovic T, et al., 2017). In addition, we replicated validation experiments on knockout mice.

6) This reviewer has had much experience evaluating and validating many of these supposedly anti-DRD2 antibodies that are cited quite commonly in the literature including AB5084P which was originally provided by Chemicon. Many of these antibodies fail to detect bands that are specific to D2DR even when the protein is transiently expressed at high levels in cell lines, i.e. there are no bands detected that are unique to cells overexpressing D2RD2 compared to untransfected cells. The problem becomes even more severe when trying to detect DRD2 endogenously expressed in brain. DRD2, in particular among GPCRs, is expressed at very low levels in brain and highly insoluble in detergents. Furthermore, even under highly denaturing conditions with urea and SDS in the sample buffer much of the DRD2 signal fails to penetrate into the PAGE gel.

RE:

We agree that DRD2 is expressed at low level in total brain, however in our work we specifically analyzed the striatum, a region receiving an extensive dopaminergic input from the *Substantia Nigra pars compacta*, and particularly enriched in dopaminergic DRD1 and DRD2 receptors.

We also agree with the Reviewer, that a portion of DRD2 is compartmentalized into the detergent-resistant fraction of the plasma membrane. Anyway, as shown by the work of Celver and Kooroor, and confirmed in our own experiments (former and revised Figure 2), DRD2 can be extracted from the detergent resistant fraction by utilizing TritonX-100.

7) Furthermore, this reviewer has attempted to validate these commercial antibodies, including AB5084P, using brain tissue from D2RD2 knockout mice. While multiple protein bands are detected, some corresponding to the predicted molecular weight of DRD2, bands specific to wild-type mice that are not present in the knockout mouse tissue are never detected.

RE:

During the course of the present work, we used lots # 1967314, 2019762, 2067217, and lot # 2279497 (giving a fainter signal) of Millipore AB5084P rabbit polyclonal antibody against DRD2. For antibody validation on DRD2 knockout mice we used residual of lot # 2279497. Each gave the same band at ~ 63 kDa in striatal mouse samples (Wang and Pickel, 2002; Rajput et al., 2009). However, in our experience the quality of different lots of any antibody may change significantly.

8) The authors of the above-cited manuscript, "How reliable are G-protein-coupled receptor antibodies?" indicate that data from GPCR antibodies should only be considered valid if the following can be demonstrated: a) disappearance of staining in knock-out animals of the target receptor, b) reduction of staining upon knock-down approaches such as siRNA treatment. This reviewer concurs and further believes that such validation should be compulsory if investigators are trying to detect and quantify endogenously expressed DRD2. Many anti-GPCR antibodies can recognize the target GPCR when it is overexpressed in cell line but only exhibit non-specific staining in vivo. If signal from these antibodies is sufficiently amplified most polyclonal antibodies will identify protein bands close to any molecular weight. In addition in immunohistochemistry experiments the antibody will show a pattern of staining but much or all of that staining could be nonspecific.

RE:

Please see point 2.

9) b) In their immunoblots the investigators show DRD2 running as a clean single band. This reviewer has never observed DRD2 running in this manner. The reviewer has worked with an anti-DRD2 antibody was validated against DRD2 knockout tissue and DRD2 from brain runs as multiple bands starting at around 60kD as the lowest and then extending to multiple higher bands. Moreover there is strong smeared signal between the more prominent bands that extends into the stacking gel.

RE:

We agree with the reviewer on this delicate issue and are definitely aware that DRD2 from brain runs as multiple bands starting at around 60kDa as the lowest and then extending to multiple higher bands. Indeed in Figure 1A3 of the manuscript a representative DRD2 immunoblotting showed multiple bands at high molecular weight.

In fact, DRD2s exist in two isoforms, that are differentially glycosylated and may be present in three post-translational states: a newly synthesized protein (45 kDa), partially glycosylated products, and a fully glycosylated mature 70 kDa receptor. Additionally, dimers can be detected. The processing to the mature receptor differs between the two isoforms, regarding timing and proportion of mature/immature products, thus justifying the variability, in presence and intensity, of the bands in different brain areas, overexpressing systems, experimental settings (Fishburn et al., 1995). The difference of smeared signal plausibly depends on different experimental conditions, such as the amount of protein loaded/different lysis buffers/running and blotting conditions, etc..

10)

In summary, while the anti-DRD2 antibody is referenced in multiple publications n DRD2 there is no proof that it recognizing DRD2, especially endogenously expressed DRD2.

RE:

We hope that our new experiment on DRD2 knockout mice, together with new radioligand binding data included in revised Figure 1, convincingly addressed this issue.

11)

The lack of validation extends to some of the other antibodies used by the investigators. The investigators need to clearly describe how the antibodies were validated. Simply showing that the antibodies detect a band running at the right molecular weight is not sufficient. The authors need to provide documentation (their own or from the supplier) showing that the antibody detects the appropriate signal in tissue known to express the target protein and that there is no signal in tissue that does not express the protein.

RE:

The commercial goat anti-RGS9-2 antibody (M20, sc-8142, Santa Cruz Biotechnology) has been previously validated in RGS9 knockout mice (Mancuso et al., J. Neurochem. 2010; see Appendix Table 1).

Unfortunately, knockout mice have not been generated for most of the several proteins evaluated in our study. According to the reviewer's suggestion, we implemented the documentation provided in the Appendix (Table 1).

12)

Alternatively, multiple antibodies targeting different epitopes of the target protein should be evaluated to ensure that the similar results are obtained with each of the different antibodies.

RE:

Unfortunately for most of the several proteins evaluated in our study multiple antibodies are not commercially available.

Referee #2 (Remarks for Author):

The article by Bonsi et al. investigates on the possible mechanisms responsible for the reduction of D2 receptor signaling in Dystonia using Dyt1 mice. Authors report that the reduction depends on increased lysosomal degradation of the D2 receptor due to reduction of proteins interacting with it. In particular, they report on the critical interaction between D2/RGS9-2 proteins, which regulates the expression and localization of these proteins. They also show that viral expression of RGS9-2 can rescue D2 functions in ChIs, as shown using an electrophysiological approach. Overall the findings are interesting and have translational value. Nevertheless, there are points that need to be addressed in support of their data and conclusions. In particular, D2 reduction is not convincing;

additional approaches should be used to support western blot analyses. Furthermore, the antibody against D2 needs to be verified using knockout mice.

RE:

We thank the Reviewer for her/his interest in our work. As discussed above (point 2-Reviewer 1), there are some considerations to take into account about the involvement of DRD2 dysregulation in DYT1 dystonia. Indeed, a reduction of DRD2 receptor in DYT1 dystonia has been consistently reported by different groups with diverse methods, in humans as well as mouse models. In human DYT1 mutation carriers, DRD2 binding is reduced by 15% (Asanuma et al., 2005). DRD2 binding has been found similarly reduced in two DYT1 mouse models, a striatal-specific torsinA knockout and a knockin for the torsinA mutation (Yokoi et al., 2011; Dang et al., 2012). Accordingly, DRD2 protein expression levels have been previously found reduced both by our group and by others in different mouse models: transgenic mice overexpressing human mutant torsinA (Napolitano et al., 2010) and mice knockin for the torsinA mutation (Dang et al., 2012).

Specific points:

1) The Dyt1 Tor1A mutation and δ gag mutation are not introduced nor it is explained how the mice were made and why only the heterozygotes are used. Readers not familiar with the disorder need this information.

RE:

We apologize for this inaccuracy and thank the Reviewer for her/his suggestion. We modified the Introduction accordingly (pages 3-4).

The Dyt1 Tor1a^{Agag} mutation responsible for almost all DYT1 dystonia cases is inherited in an autosomal dominant manner, and symptomatic patients are heterozygous for the mutation (see: <http://www.mdsgene.org/d/4/g/14>). Homozygous Dyt1 Tor1a^{Agag/Agag} and Dyt1 Tor1a^{-/-} mice die 1-3 days after birth.

2) In Fig 1 all the experiments shown are made with extracts from P60 mice while in the text is mentioned P60-P90. What was done at P90

RE:

We unreservedly apologize for the lack of clarity; the figure legend has been amended. The text mentioning the range P60-P90 (Results - page 4, line 3 and page 6 line 2) refers to the immunoblotting analysis performed on adult mice reported in Figure 1A1 and Figure 1B, respectively. Figure 1A2, A3, A4 show representative immunoblottings performed at the indicated postnatal ages.

3) The D2 and RGS9-2 down-regulation is not convincing; less exposed western blot might be more convincing. What is the size of D2 on these western blots? Most commercial antibodies against D2 are not specific, it would be good to run an extract from D2KO mice as control of antibody specificity. Also additional experiments should be made in support of this important point. For instance, D2 ligand binding experiments on striatal membrane would be very appropriate.

RE:

According to the Reviewer's suggestion we replaced the immunoblots with less exposed ones in revised Figure 1. In accordance with the literature (Wang and Pickel, 2002; Rajput et al., 2009), in our western blots striatal DRD2 runs at ~63kDa.

As the Reviewer may see above (please see point 2 of the response to Reviewer 1), we run western blotting and immunohistochemistry experiments with the AB5084P antibody on DRD2 knockout samples kindly provided by Prof. Emiliana Borrelli. These experiments show lack of DRD2 signal in knockout samples.

Additionally, as suggested by the Reviewer, we also performed radioligand binding experiments on striatal sections with 3H-spiperone. These data (included in revised Figure 1) show a significant reduction of DRD2 binding in the striatum of mutant mice, with respect to wild-type littermates, supporting western blotting data.

4) Similarly, if RGS9-2 is responsible for the D2 trafficking in the cells and since it is very hard to evaluate this *in vivo*, *in vitro* experiments should be performed using transfected D2 and RGS9-2 in mammalian cells. In this system it would be possible to change the ratio of the two proteins and analyze more in depth the lysosomal degradation.

RE:

We agree with the Reviewer, this would be undoubtedly a fruitful approach. In fact, we tried to set up such experiments. However, DRD2 overexpression *per se* has been shown to affect RGS9-2 compartmentalization (Cervera et al., 2012). Furthermore, our main interest in the present work, was to investigate the changes induced by loss-of-function *DYT1* mutation affecting torsinA level and function. Thus, to replicate the *DYT1* condition, we need to downregulate torsinA in a heterologous system overexpressing both DRD2 and RGS9-2. The three actors multiply the experimental conditions, and for that reason they would be matter of further investigation.

5) Is D2 expression, in the brain of mice heterozygous for any of the proteins mentioned in the autophagy-lysosomal pathway, reduced? This would support results contained in fig. 3-6, where at present differences between WT and KO are not striking.

RE:

We looked with interest for such mouse models. Unfortunately we did not find available commercial animal models for these proteins; furthermore, obtaining the Italian Ministry permission to utilize a different mouse strain requires ~6 months.

Referee #3 (Remarks for Author):

This is an interesting study that provides insight into mechanisms underlying an animal model of severe DYT1 dystonia, the TOR1A mouse. They discovered reduced striatal expression of dopamine receptor 2 (DRD2) in these mice and propose that reduced expression of proteins that counteract DRD2 internalization and degradation mechanisms (RGS9-2 and spinophilin). As DYT-1 dystonia patients respond poorly to dopaminergic drugs, these findings have significant implications for pathophysiology and treatment. Overall the data are convincing and the paper well-written. There are several issues that the authors should address prior to publication.

RE:

We thank the Reviewer for her/his appreciation of our work.

Major comments:

1. Please provide some background on the TOR1A and TOR1A(deltaGAG) mutant mouse models, the functions torsinA, and its relationship to DRD2 signaling and dystonia.

RE:

We apologize for this inaccuracy and thank the Reviewer for the suggestion. We modified the Introduction accordingly (pages 3-4).

2. Figure 1 The reductions in RGS9-2 levels in the TOR1A mouse are not impressive. It would be important to know if its obligate binding partners Gb5 or R7BP are concomitantly reduced in these experiments.

RE:

We agree with the Reviewer. Accordingly, we performed new immunoblotting experiments (shown in revised Figure 2a1; and discussed in the Results section, page 6) to measure concomitantly in the same experiment the level of protein expression of RGS9-2 and its binding partners Gb5 and R7BP from each sample of total striatal lysate.

3. Figure 2. What is meant exactly by a "trend" in the reduction of DRD2 in DRMs of TOR1A striatum? Since the graph shows a wide range of DRD2 values (actually 2 distinct groups) in the +/- mutant mice, a few more mice could determine whether this is truly significant or not. Since RGS9-2 is unstable without Gb5 or R7BP, is the localization of these proteins in DRMs similarly altered?

RE:

We thank the Reviewer for this suggestion. We performed additional immunoblotting experiments quantifying DRD2, Gb5 and R7BP in the DRM. The results are shown in revised Figure 2c1,c2,c3,c4 and discussed at page 7-8 of the Results. The results show that DRD2 is significantly reduced also in the DRM, while R7BP is increased in this compartment, in parallel with RGS9-2. Of interest, Gb5 is unchanged both in total lysate and in the DRM.

4. Figure 3. What is meant by "sham" transfection and why wasn't a control (nontargeting) shRNA not used? Figure 3e: The "trend" referred to is not apparent; either the data are significant or not-please clarify.

RE:

We agree with the Reviewer, the use of a non-targeting shRNA is required in silencing experiments. Indeed, before using them *in vivo*, we tested the shRNA *in vitro*, to assess their efficacy by western blotting and immunocytochemistry. For these assays, we utilized HEK293T-RGS9 cell clones, derived by HEK293T cells transfected with a RGS9-mCherry viral construct to permanently express RGS9. As shown in the figure below, lysates of HEK293T-RGS9, transfected with control shRNA CMV-GFP, or shRNA-34 and shRNA-35. We consistently observed reduction of RGS9 level in cells transfected with either shRNA-34 or shRNA-35, and no effect of control shRNA. The immunocytochemistry images show a HEK293T-RGS9 cell (mCherry) showing GFP signal (green), and lack of RGS9 immunolabeling (blue) after silenced by shRNA-34. In consideration of the results of the *in vitro* validation, *in vivo* we used shRNA-34 and shRNA-35, and injection of saline in the contralateral side as control ("sham").

The sentence reporting a "trend" was corrected.

A.**B.**
Minor comments:

Page 3, last paragraph "These findings can [explain] the paradox

RE:

Corrected as suggested.

2nd Editorial Decision

6 November 2018

Thank you for the submission of your revised manuscript to EMBO Molecular Medicine. We have now received the enclosed reports from the referees that were asked to re-assess it. As you will see the reviewers are now globally supportive and I am pleased to inform you that we will be able to accept your manuscript pending minor editorial amendments including a response to referee 1. It seems important to us that you comply with ref. 1 request to provide and show in all WB the entire D2DR lane. Please discuss the rest of the report. As for raw data, please see [our guidelines].

***** Reviewer's comments *****

Referee #1 (Remarks for Author):

The authors have responded to criticism about the specificity about the anti-D2DR antibody that was used in their study by providing additional control data with D2DR knockout mouse tissue to support their position that their antibody, Millipore AB5084P, was indeed recognizing D2DR in the brain. However, this reviewer remains somewhat skeptical about whether the original immunoblotting experiments reported in the paper were indeed measuring brain D2DR.

For example, the new control immunoblot in their "Point-by-Point Response" document comparing immunoblots of D2DR vs wild-type clearly shows a streak of D2DR signal extending up into molecular weight range over 95 kDa. The streaking signal is relatively bright compared to the single brightest band at 62 kD. However, the representative blots shown Figure 1A3 show a completely different banding pattern. The authors argue that there could be experiment to experiment variation in the banding pattern of D2DR on an immunoblot. However, the D2DR signals in the representative figures in the paper appear to be quite consistent.

If significant signal from D2DR is found within a streak and not only in a single band, then it is important that the authors show the entire D2DR lane in all of their representative blots. Furthermore, because the specificity of the D2DR signal is an important determinant for the validity of the conclusions drawn in the manuscript it would be helpful if the authors can provide all of the raw images of the D2DR immunoblots that were utilized to quantify D2DR expression levels in each of the figures.

Also, in the rebuttal the total integrated D2DR signal in the streak above 62 kDa is likely larger than that in the band at 62 kD. Therefore, it is not clear what area of the immunoblot the author were isolating and quantifying as D2DR signal. The variation in the banding pattern shown in the rebuttal vs what is shown in the manuscript raises some skepticism about how valid or significant 15-20% alterations in D2DR signal between experimental and control samples might be.

In the rebuttal the authors indicate that Clever and Kavoov were able to extract D2DR signal from DRM fraction using triton X100. This reviewer carefully studied the Methods section in that paper by Clever et al and it is indicated that the D2DR signal in the DRM was extracted by solubilizing the tritonX100 insoluble fraction in SDS-containing buffer. This would make sense, as by definition, detergent-resistant membrane fraction proteins are insoluble in milder detergents such as triton.

As a consequence of the above inconsistencies there remain significant concerns about the conclusions drawn by the authors.

Referee #3 (Comments on Novelty/Model System for Author):

Adequate statistics and methodology

Referee #3 (Remarks for Author):

The authors have improved the manuscript and addressed my previous concerns.

2nd Revision - authors' response

21 November 2018

Referee #1 (Remarks for Author):

The authors have responded to criticism about the specificity about the anti-D2DR antibody that was used in their study by providing additional control data with D2DR knockout mouse tissue to support their position that their antibody, Millipore AB5084P, was indeed recognizing D2DR in the brain. However, this reviewer remains somewhat skeptical about whether the original immunoblotting experiments reported in the paper were indeed measuring brain D2DR.

RE:

We regret that the Reviewer is not fully convinced despite the additional immunoblotting and confocal microscope experiments on DRD2 knockout mice showed lack of immunolabeling, and further radioligand binding assay demonstrated similar reduction of DRD2 in our mouse model.

For example, the new control immunoblot in their "Point-by-Point Response" document comparing immunoblots of D2DR vs wild-type clearly shows a streak of D2DR signal extending up into molecular weight range over 95 kDa. The streaking signal is relatively bright compared to the single brightest band at 62 kD. However, the representative blots shown Figure 1A3 show a completely different banding pattern. The authors argue that there could be experiment to experiment variation in the banding pattern of D2DR on an immunoblot. However, the D2DR signals in the representative figures in the paper appear to be quite consistent.

If significant signal from D2DR is found within a streak and not only in a single band, then it is important that the authors show the entire D2DR lane in all of their representative blots. Furthermore, because the specificity of the D2DR signal is an important determinant for the validity of the conclusions drawn in the manuscript it would be helpful if the authors can provide all of the raw images of the D2DR immunoblots that were utilized to quantify D2DR expression levels in each of the figures.

Also, in the rebuttal the total integrated D2DR signal in the streak above 62 kDa is likely larger than that in the band at 62 kD. Therefore, it is not clear what area of the immunoblot the author were isolating and quantifying as D2DR signal. The variation in the banding pattern shown in the rebuttal vs what is shown in the manuscript raises some skepticism about how valid or significant 15-20% alterations in D2DR signal between experimental and control samples might be.

RE:

In accordance with previous work by others, that identified a 63kDa band as DRD2 in the striatum, and with our present validation in knockout mice, we quantified the clear 63kDa band in our immunoblots. We observed some additional clear bands at higher molecular weight in young mice (from P7 to P21, see Figure 1A3), but not in adult mice.

We are convinced that some differences in DRD2 signal in our experiments depend on experimental conditions, such as amount of loaded proteins, exposure time, antibody dilution. For example, to assure detection of possible light bands in the knockout sample, 40 micrograms of proteins from knockout and control samples were loaded in the gel shown in the rebuttal letter. As the Reviewer correctly pointed out, this immunoblot shows a smear signal above the clear band detected at ~63kDa. Novel Figure EV1 shows an immunoblot of a gel loaded with 30 micrograms of proteins. However, most immunoblots were run loading 15-20 micrograms of proteins and smear signal was not detected, as shown by uncropped gels images provided as Source Data for each figure. Unfortunately it was not possible to provide raw images of all the immunoblots from which data were obtained to prepare the manuscript, considered the high number of replicates included in the quantification graphs.

In the rebuttal the authors indicate that Clever and Kavour were able to extract D2DR signal from DRM fraction using triton X100. This reviewer carefully studied the Methods section in that paper by Clever et al and it is indicated that the D2DR signal in the D2DR was extracted by solubilizing the tritonX100 insoluble fraction in SDS-containing buffer. This would make sense, as by definition, detergent-resistant membrane fraction proteins are insoluble in milder detergents such as triton.

RE:

We apologize for this inattention in the rebuttal letter. As described in the Methods section, we used the method described by Celver and Kovoov, who utilized TritonX-100 to separate detergent-soluble vs. detergent-resistant membrane fractions.

Referee #3 (Comments on Novelty/Model System for Author):

Adequate statistics and methodology

Referee #3 (Remarks for Author):

The authors have improved the manuscript and addressed my previous concerns.

RE:

We would like to thank the Reviewer for providing insightful comments contributing to the improvement of the manuscript.

Corresponding Author Name: ANTONIO PISANI
Journal Submitted to: EMBO MOLECULAR MEDICINE
Manuscript Number: EMM-2018-09283